# Immune cellular networks underlying recovery from influenza virus infection in acute hospitalized patients

Thi H. O. Nguyen [1], Marios Koutsakos[1], Carolien E. van de Sandt [1,2], Jeremy Chase Crawford [3],
Liyen Loh [1], Sneha Sant[1], Ludivine Grzelak[4], Emma K. Allen[3], Tim Brahm [3], E. Bridie Clemens[1],
Maria Auladell [1], Luca Hensen [1], Zhongfang Wang[1], Simone Nüssing[1], Xiaoxiao Jia[1], Patrick Günther [1],
Adam K. Wheatley [1], Stephen J. Kent [1,5,6], Malet Aban[7], Yi-Mo Deng[7], Karen L. Laurie[7], Aeron C. Hurt[7],
Stephanie Gras[8,9,10], Jamie Rossjohn [8,9,11], Jane Crowe[12], Jianqing Xu [13], David Jackson [1],
Lorena E. Brown[1], Nicole La Gruta [8], Weisan Chen [10], Peter C. Doherty [1], Stephen J. Turner [14],
Tom C. Kotsimbos[15,16], Paul G. Thomas [3], Allen C. Cheng[17,18,19 ✉] & Katherine Kedzierska [1,19 ✉]

How innate and adaptive immune responses work in concert to resolve influenza disease is
yet to be fully investigated in one single study. Here, we utilize longitudinal samples from
patients hospitalized with acute influenza to understand these immune responses. We report
the dynamics of 18 important immune parameters, related to clinical, genetic and virological
factors, in influenza patients across different severity levels. Influenza disease correlates with
increases in IL-6/IL-8/MIP-1α/β cytokines and lower antibody responses. Robust activation
of circulating T follicular helper cells correlates with peak antibody-secreting cells and
influenza heamaglutinin-specific memory B-cell numbers, which phenotypically differs from
vaccination-induced B-cell responses. Numbers of influenza-specific CD8[+] or CD4[+] T cells
increase early in disease and retain an activated phenotype during patient recovery. We
report the characterisation of immune cellular networks underlying recovery from influenza
infection which are highly relevant to other infectious diseases.

A full list of author affiliations appears at the end of the paper.

Millions of people are hospitalized with severe influenza disease annually. In 2017, an estimated 9.5 million people globally were hospitalized with influenza virus lower respiratory tract infections, for a total of 81.5 million days[1]. Moreover, an estimated 243,000–645,000 of seasonal influenza-associated respiratory deaths occur annually[2]. Although seasonal influenza virus infections can cause debilitating illness leading to hospitalization and death[3], viral, host and immune factors determining the disease severity are unclear, as are the precise mechanisms of why some individuals present with a mild 'asymptomatic' infection, while others, including previously healthy or vaccinated individuals, succumb to severe viral pneumonia and fatal influenza disease. Increased susceptibility to influenza virus infection and exacerbation of disease severity can reflect an overactivation of the innate immune system, leading to hypercytokinemia, alveolar oedema and pulmonary complications, and/or impaired humoral and cellular immunity, delaying the recovery phase. While neutralizing antibodies can reduce disease transmission and viral load, the current vaccination regimens targeting antibody responses[4] are often short-lived and directed against specific influenza strains, thus not protective against unpredicted influenza viruses.

Based on human studies of natural and experimental influenza virus infection, it is well-established that in the absence of neutralizing antibodies, pre-existing memory CD8+ T cells and CD4+ T cells can reduce disease severity[5–14]. Published evidence also associates genetic host factors, such as specific HLA types and interferon-induced transmembrane protein 3 (IFITM3) single-nucleotide polymorphisms (SNPs) with clinically poor outcomes[15–19]. The IFITM3 rs12252-C/C genotype along with early detection of inflammatory mediators IL-6, IL-8, IL-10 and MIP-1β measured in the blood within 48 h of hospital admission, were associated with fatal disease outcomes in our 2013 study of hospitalized H7N9-infected patients in China ($n = 18$ patients, 12 recovered, 6 died)[18]. Association of hypercytokinemia (IL-8, MCP-1, IP-10) with fatal outcomes was first reported for the H5N1 virus[20]. In contrast, recovery from severe H7N9 infection was associated with lower cytokine levels and prominent CD8+ T cell responses[18]. Inflammatory serum cytokines IL-6 and IL-8 have also been linked to severe seasonal and pandemic influenza virus infections in hospitalized adult patients[21] and predicted hospitalization in a household cohort of naturally-infected individuals, including infants, children and adults[22]. In critically-ill children, higher levels of blood cytokines (IL-6, IL-8, IP-10, GM-CSF, MCP-1, and MIP-1α) measured within 72 h of ICU admission correlated with fatal influenza virus infection[23].

There is scant data on the main drivers of severe influenza resulting in hospitalization or death[24,25]. Previous studies of influenza-infected patient cohorts mainly focused on a very select number of immunological parameters, with only three studies combining innate (cytokines/monocytes/NK cells) and adaptive T cell immunity[12,26,27], and two other studies incorporating innate immunity, T cell immunity and antibody responses[10,28]. The limited number of parameters measured is often due to a difficulty in obtaining longitudinal samples from acutely-infected patients combined with the amount of biological material required per assay. Only one other study, the SHIVERS report[28] analyzed convalescent samples 2 weeks post-enrolment (along with one acute sample) and showed prolonged immune activation of innate and adaptive cellular responses in hospitalized patients compared to non-hospitalized patients. Thus, there is a clear deficiency in our understanding of the specific interplay between genetics, innate and adaptive immune responses driving recovery from acute viral pneumonia.

Here, we utilize longitudinal samples obtained from patients hospitalized with acute influenza disease to elucidate how innate and adaptive immune responses function together to resolve influenza disease. We report, at a very high level of resolution, the overall breadth and kinetics of 18 key immune parameters: the antiviral/inflammatory cytokines and chemokines, hemagglutinin (HA)-directed antibodies, HA-probe-specific B cells, antibody-secreting cells (ASCs), circulatory CD4+ T follicular helper (cTfh) cells, influenza peptide/MHC-specific CD8+ and CD4+ T cells, IFN-γ-producing CD8+ T cells, CD4+ T cells, natural killer (NK) cells, mucosal-associated invariant T (MAIT) and γδ T cells, and granzymes A, B, K, M and perforin expression in CD8+, CD4+, NK and MAIT cells. This comprehensive panel of immune parameters, combined with host genetic factors and patient clinical data, enables a detailed dissection of human factors driving susceptibility, severity and recovery in patients hospitalized with seasonal influenza viruses, highly relevant to other infectious diseases, especially newly-emerging respiratory infections.

## Results

**Patient DISI cohort.** We recruited 64 patients admitted to the Alfred Hospital (Prahran, Australia) between 2014 and 2017 into our "Dissection of Influenza-Specific Immunity" (DISI) cohort. The inclusion criteria for the DISI study included hospital admission of consenting adult patients with influenza-like illness (ILI). The longitudinal study involved serial blood and nasal swab samples collected from influenza-PCR-positive patients (influenza+, $n = 44$) within 1–2 days of hospital admission to discharge and follow-up blood samples ~30 days later, allowing analyses of the recovery phase (Fig. 1a). Patients who were influenza-PCR-negative (influenza-, $n = 20$) were included, which represented a unique set of patients with other respiratory illnesses, some caused by viral infections. The DISI cohort included one death patient in each of the influenza+ and influenza- groups (#11 and #28, respectively). The remaining cohort was admitted to the Respiratory/General Ward with one influenza+ patient requiring ICU (#62). 53 patients presented with ILI, while 6 influenza+ and 1 influenza- patient presented with pneumonia (Supplementary Table 1, Supplementary Data 1).

We successfully recalled 80% (35/44) of influenza+ patients at a median of 41 days after disease onset and 75% of our influenza- patients (15/20, median 39 days). Overall, influenza+ patients were predominantly (55%) infected with the H3N2 influenza A virus (IAV) subtype (Fig. 1b and Supplementary Table 1), followed by the co-circulating IAV 2009 pandemic H1N1 (pH1N1)-like strain (16%) and two co-circulating influenza B virus (IBV) strains (Phuket/3073/2013 (9%) and Brisbane/60/2008 (7%)), representing the dominant strains for each year in Australia. IAV-infected patients (median 58 yrs, $n = 34$) were significantly older than IBV-infected (45 yrs, $n = 10$) and influenza- patients (47 yrs, $n = 20$) (Fig. 1c). Apart from age, patient demographics in influenza+ and influenza- groups were well-matched, with 86% and 70%, respectively, having one or more high-risk conditions for severe influenza disease. Both groups had a median of 4 days in hospital (Supplementary Table 1, Supplementary Data 1). Time in hospital was significantly lower for seasonal influenza+ patients compared to the more severely-ill H7N9 cohort[18] with a median 14 days for recovery ($n = 12$, $p < 0.0001$, Kruskal–Wallis test) and a median of 33 days for those who died ($n = 6$, $p = 0.0437$, Kruskal–Wallis test) (Fig. 1d). The lower hospital burden reflects the lower severity of seasonal influenza compared to avian A/H7N9 disease.

**Cytokine clusters positively correlate with each other during influenza virus infection.** Hypercytokinemia of early inflammatory mediators (IL-6, IL-8, IL-10, MCP-1, MIP-1α, and MIP-1β) have been associated with an increase in disease severity causing death in

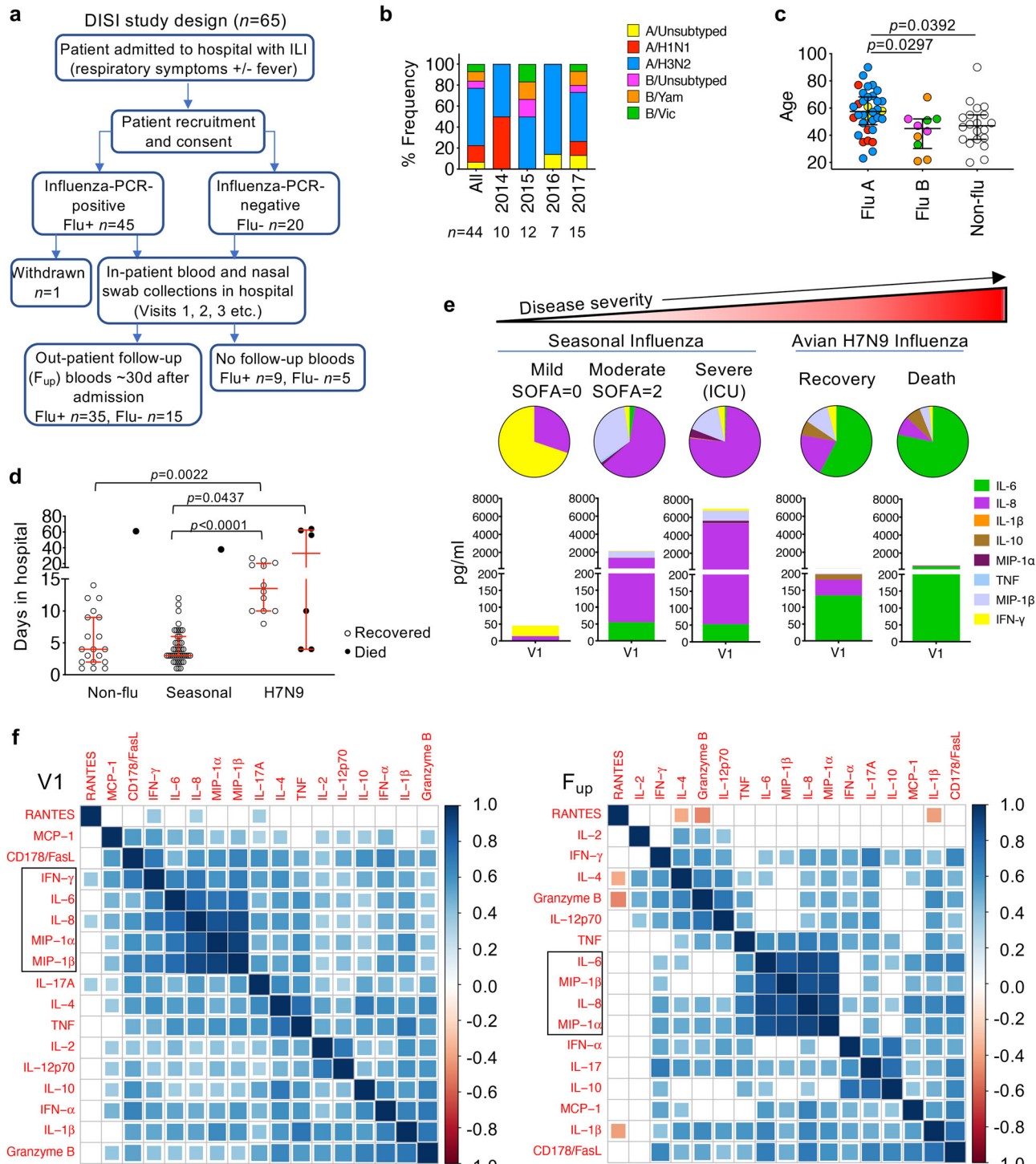

**Fig. 1 Cluster analysis of patient clinical information, genetic characteristics and inflammatory cytokines. a** Flow chart of study design. **b** Frequency of patients infected with seasonal influenza virus strains. *N* values are below for total and for each year. **c** Age distribution by influenza A (3 unsubtyped, 7 H1N1, 24 H3N2), influenza B (3 unsubtyped, 4 YAM, 3 VIC) and influenza-negative (Non-flu, Flu-) patients (*n* = 20). Virus strain colors match those in (**b**). **d** Days in hospital for DISI cohort and H7N9 cohort. **c**, **d** Bars indicate the median and IQR, statistical significance (0.0001 > *p* < 0.05) was determined using the Kruskal–Wallis test (two-tailed). **e** Representative serum levels and distribution of pro-inflammatory cytokines in patients, measured within the first 2–3 days of hospital admission (Visit 1, V1), with varying disease severity. **f** Partial correlation plots showing the degree of correlation between every chemokine/cytokine pair in influenza+ patients at V1 and F_up. The color corresponds to the correlation coefficient and the size of the colored squares correspond to the FDR-adjusted *p* value. Correlations that are not significant (*p* > 0.05) result in white boxes.

a number of hospitalized cohorts[18,21–23]. Pro-inflammatory cytokines and chemokines measured in the blood of acute representative influenza+ patients showed increased cytokine levels, particularly IL-6 and IL-8, between mild and moderate patients as defined by lower (0–1) or higher (2–6) sequential (sepsis-related) organ failure assessment (SOFA) scores[29], respectively, and in the more severe patient requiring ICU support (Fig. 1e). Heightened IL-6 levels were also associated with the more severe 2013 H7N9-infected patients (Fig. 1e)[18]. After adjusting for age and the sampling time after days of disease onset, IL-6, IL-8, MIP-1α, MIP-1β and IFN-γ were significantly, strongly positively correlated with each other early after infection at V1 timepoint. This inter-correlated cytokine cluster except IFN-γ was still strongly present at convalescence, as well as the convergence of a IFN-α, IL-17 and IL-10 cluster (Fig. 1f). Therefore, our data support previous reports that IL-6, IL-8, MIP1α and MIP-1β serve as biomarkers for severe influeza disease.

**Antibody responses increase in magnitude and breadth at convalescence.** The annual influenza vaccine is provided for free for high-risk groups under the National Immunization Program. Despite the high proportion of influenza+ patients in this cohort being defined as high-risk due to pre-exisiting comorbidities (86%), only 48% of these patients were vaccinated against influenza prior to infection, compared to 65% of influenza- patients (Supplementary Table 1). There was rapid antigenic drifting of the H3N2 viruses between 2014 and 2017, with more annual updates of the H3-HA vaccine component compared to the less variable H1-HA and B-HA components (Supplementary Table 2). Rapid drift of the H3N2 component was evident in the viral HA-sequence analyses of nasal swabs from H3N2-infected patients with sufficient RNA. All 12 H3N2 viruses isolated from patients, including 9 who had been vaccinated, were from a different H3N2 clade when compared to the current H3N2 seasonal vaccine strain (Fig. 2a). Conversely, fewer H1-HA and B-HA sequence variations were observed between isolates from the infected patients compared to the WHO reference strains (Supplementary Fig. 1). Interestingly, despite the H3N2 clade-mismatches, all H3N2-infected patients (except one A/unsubtyped patient) generated HA inhibition (HAI) antibody titres of 40 and above (equivalent to $\log_2(HAI/10)\geq2$) against the H3 vaccine strain (same year as infection) at follow-up, compared to 54% at acute timepoints (Fig. 2b), suggesting a boost in strain-specific antibodies that may have contributed to the significant increase in vaccine-strain titres ($p < 0.0001$, Mann–Whitney test). However, only marginal increases in antibody titres were observed in H1- and B-infected patients at follow-up (Fig. 2b). Geometric mean titres (GMT) were low for all three H1, H3 and B subtypes during acute infection, but their ability to mount antibody responses at follow-up was generally comparable to vaccine-induced responses in a healthy cohort but with greater GMT fold-changes (Fig. 2c, d). As expected, antibody titres were low and remained unchanged in the influenza-group (Fig. 2c, d).

Previous exposures to past influenza strains, influenced by age, can affect our immune responses to current influenza virus infections[30]. To study the breadth of antibody responses, HAI antibody landscapes were generated against current and older influenza strains from the last century that the patients may have previously encountered (Supplementary Table 2). Antibody responses to past strains were minimal during acute infection (V1 and V2) (Fig. 2e, f). However, at follow-up, influenza+ patients induced robust antibody responses to older strains from the same subtype that they were infected with, a phenomenon known as back-boosting[30], but not against the other subtypes (Fig. 2e, f). Therefore, compared to the influenza- and healthy cohorts, influenza+ patients started with lower antibody titres at

hospitalization, perhaps due to lower vaccine coverage, but were able to mount robust and broad antibody responses at convalescence.

**Activation of cTfh, ASCs and influenza-specific memory B cells prior to recovery.** Tfh cells are essential for generating high-affinity memory B cells in germinal centers (GCs) and are located in secondary lymphoid organs[31]. cTfh cells share phenotypic and functional properties to GC Tfh cells, and it has been shown that inactivated influenza vaccination (IIV) induces expansion and PD-1/ICOS-activation of CD4+CXCR5+CXCR3+ cTfh type-1 (cTfh1) cells, which correlated with antibody and CD19+CD27++CD38++ ASC responses[4,32].

Here, we show that activated PD-1+ICOS+ cTfh1 cells emerged within influenza-infected patients in parallel with ASC responses, both peaking between days 7 and 10 after disease onset in both influenza+ and influenza- groups, prior to patients' recovery (Fig. 3a–d, Supplementary Fig. 2a). Activated cTfh1 cells peaked higher in influenza+ patients compared to influenza-groups, as shown by the separation of the 95% confidence intervals and individual patients' dynamic responses (Fig. 3c-d, Supplementary Fig. 2a). The number of ASCs was significantly higher (~2–8 median fold) during acute infection compared to follow-up for both influenza+ and influenza- groups (Fig. 3e). Moreover, influenza+ patients at acute phase had higher ASC numbers than our previously described healthy cohort prior to vaccination (4.4 median fold) and at d7 post vaccination (2.6 median fold), the peak of the vaccine response. Activated cTfh1 cells were also trending higher at acute compared to follow-up (~2 fold, $p = 0.0519$, Mann–Whitney test, Fig. 3f). The number of activated cTfh1 cells for acute influenza+ patients was only 1.5 median fold higher compared to healthy individuals at baseline, and was slightly lower than d7 post vaccination in the healthies. This could be in part due to the patients' lymphopenic state during acute illness where the total CD45+ lymphocyte and subset-specific cell counts were significantly lower at acute-V1 compared to follow-up, except for B cells, which explains the higher ASC numbers observed in acute patients. Cell counts were less variable in the influenza- patients and more stable in healthy donors (Supplementary Fig. 3a–c).

Nevertheless, numbers of activated cTfh1 cells in influenza+ patients were significantly higher than the cTfh2 and cTfh17 subsets, at both acute and follow-up timepoints (Fig. 3f). Further, cTfh1 responses strongly correlated with ASC responses ($r_s = 0.7060$, $p < 0.0001$, Spearman test) during acute influenza virus infection, but less so for total cTfh ($r_s = 0.5397$, $p = 0.0003$, Spearman test) and cTfh2 cells ($r_s = 0.3741$, $p = 0.0174$, Spearman test) and no correlation with cTfh17 subsets (Fig. 3g). Whereas strong correlations were observed between acute ASC responses and total cTfh, cTfh1 and cTfh2 T cell subsets for influenza- patients (Fig. 3h). Acute ASC responses also correlated with high antibody responses ($\log_2(HAI/10)\geq2$ or HAI≥40, Fig. 3i). However, acute cTfh responses did not significantly correlate with antibody responses, perhaps due to the low titres observed in some patients during acute infection, in contrast to a vaccination response where IIV-induced cTfh1 responses correlated with the fold-change in antibody titres (d28 over d0 baseline)[4].

To evaluate influenza-specific memory B cell responses during natural influenza virus infection in comparison to vaccination, recombinant HA probes relevant to the 2014-2017 vaccine strains were used to detect class-switched IgD− HA-specific B cells in influenza+ patients (Fig. 4a, Supplementary Fig. 2b). These rHA probes have been extensively validated for their specificity[4,33,34]. Numbers of rHA+IgD− B cells in total and per probe did not

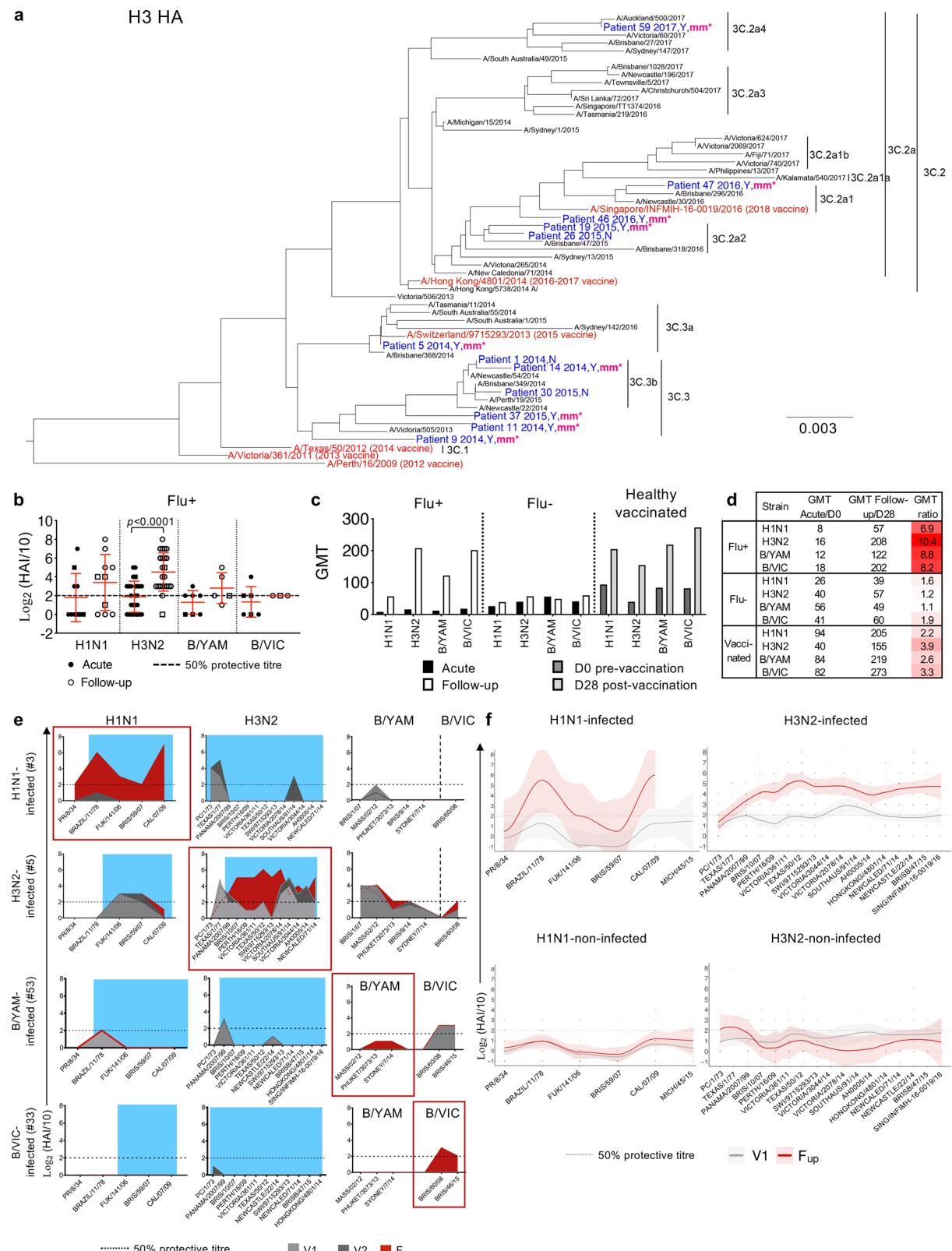

significantly change between acute and follow-up timepoints (Fig. 4b), and were comparable to the magnitudes observed following IIV[4]. Phenotype and isotype distributions of the total B cells remained stable between acute and follow-up timepoints (Fig. 4c, d). Specifically, rHA+IgD− B cells for H3 and B probes were more CD21loCD27hi activated memory in proportion at the acute timepoint compared to follow-up (H3: p = 0.0007, B: p =

0.0017, Tukey's multiple comparison test), before becoming more CD21hiCD27hi resting memory at follow-up compared to acute timepoints (H3: p = 0.0004, B: p < 0.0001, Tukey's multiple comparison test) (Fig. 3c, e top panels). Conversely, in a healthy vaccinated cohort, the resting memory phenotype was most prominent at baseline, before becoming more activated memory from days 7-28 following IIV (Fig. 3f, top panel). The isotype

**Fig. 2 Viral analysis and antibody responses. a** H3N2 phylogenetic tree of HA amino acid sequences from previous WHO reference strains in black, influenza vaccine strains in red and sequences isolated from the nasal swab of 12 H3N2-infected patients in blue. Patient number is followed by the year of recruitment, yes (Y) or no (N) for prior vaccination in the year of infection, and "mm*" indicates whether the vaccine was a clade mismatch in that year. Scale bar represents the number of substitutions per site. **b** Antibody HAI titers of influenza+ (Flu+) patients at acute (V1 or V2) and follow-up timepoints from the relevant infected strain (H1N1 $n = 10$, H3N2 $n = 26$, B/YAM $n = 7$, B/VIC $n = 6$) (mean and SD are shown). Statistical significance ($0.0001 > p < 0.05$) was determined using a two-sided Mann–Whitney test between acute and follow-up per strain. **c, d** Geometric mean titers (GMT) per strain in influenza+ and influenza- (Flu-) patients at acute and follow-up, and from a healthy vaccinated cohort at days 0 and 28 post-vaccination. **b–d** Both H1N1 and H3N2 titers are shown for three A/unsubtyped patients and both B/YAM/Phuket/3073/2013 and B/VIC/Brisbane/60/2008 titres are shown for three B/unsubtyped patients (square symbols). **e** Representative antibody landscapes from a patient infected with H1N1, H3N2, B/YAM or B/VIC virus. Blue shading indicate period of potential exposure based on the year born. **f** Antibody landscapes of H1N1- and H3N2-infected ($n = 7$ and 23, respectively) and H1N1- and H3N2-non-infected patients ($n = 45$ and 29, respectively). Lines and shading indicate the GMT and 95% confidence intervals, respectively. Gradient colored dots indicate individual titres.

distribution in influenza+ patients was enriched for $IgG^-IgM^-$ cells (mostly $IgA^+$) for H3, and $IgA^+$ cells for the B-probe at acute timepoints, before becoming largely $IgG^+$ at follow-up for each probe (Fig. 3c, e bottom panels). In contrast, the isotype distribution of $rHA^+IgD^-$ B cells did not change in healthy donors following IIV (Fig. 3f, bottom panel). Thus, our analyses show striking differences after influenza virus infection compared to vaccination of recruited $rHA^+IgD^-$ B cells at both phenotypic and isotype levels.

Although patients' HA-specific B cell responses did not positively correlate with their acute antibody responses which were generally low, there was a significant positive correlation between activated or resting memory HA-specific B cells with their acute ASC response, as well as resting memory HA-specific B cells with the acute cTfh responses (Supplementary Fig. 4). Taken together, we show prominent activation of cTfh1 cells during acute influenza virus infection, at the time of the ASC and influenza-specific memory B cell responses.

**Influenza-specific adaptive CD8+ and CD4+ T cells respond early after infection.** Our previous work showed that rapid recovery from severe A/H7N9 was associated with early IFN-γ-producing CD8+ T cell responses, while late recovery involved a network of humoral (Abs) and cellular (CD8+, CD4+, NK) responses[12]. Here, influenza+ patients' PBMCs were incubated with live seasonal H1N1, H3N2, B/YAM and B/VIC viruses to measure influenza-specific innate (NK, γδ, $CD161^+TRAV1-2^+$ or MAIT cells) and adaptive (CD4+ and CD8+ T cells) immune responses elicited via IFN-γ production after 18 h (Fig. 5a, b, Supplementary Fig. 5a). Kinetics of influenza-specific IFN-γ-producing cells across days after disease onset showed low numbers of IFN-γ-producing populations across all cell subsets at patients' admission. These however increased over the course of infection, before stabilizing at convalescence (Fig. 5c). Despite lymphopenia, a fixed number of cells were added to the assay, and therefore the ability of adaptive CD8+ and CD4+ T cells to generate IFN-γ-responses, as a frequency per subset, continued to increase during acute infection until ~15 days after disease onset (Fig. 5d). Conversely, for innate cells, their ability to produce IFN-γ responses did not change, further suggesting that adaptive T cells can increasingly respond to the virus during acute infection and are the main drivers of early recovery.

**Cytolytic potential of MAIT, NK and CD8+ T cells in influenza+ patients.** To assess the cytolytic potential within the influenza+ patients' cell subsets, we measured cytotoxic molecules: granzymes (A, B, K and M) and perforin (Supplementary Fig. 6a). Overall, MAIT cells had the highest frequency of total cytotoxic molecules expressed, followed by NK cells and CD8+ T cells, with minimal production in CD4+ T cells (Fig. 5e). However, for

MAIT cells, the influenza+ patients at acute and convalescence exhibited significantly lower levels compared to healthy donors (demographics in Supplementary Table 3), but on the contrary for NK cells, possibly suggesting differences in degranulation kinetics. No differences were observed for CD8+ and CD4+ T cell subsets. Individual expression of cytolytic molecules revealed different patterns between cell subsets and cytolytic molecules (Fig. 5f). Patient NK cells had significantly higher levels of granzymes (A, B, M) and perforin compared to healthy donors, but significantly lower levels of granzyme K. Patient MAIT cells expressed lower levels of granzyme A compared to healthy donors, while patient MAIT and CD8+ T cells had lower granzyme K and perforin expression. No significant differences were observed between acute and follow-up subsets. Furthermore, co-expression of multiple cytolytic molecules (4+) was significantly reduced in influenza+ patients, both at acute and follow-up, compared to healthy individuals across all cell subsets (NK cells: acute versus healthy $p = 0.0002$, $p < 0.0001$ for all comparisons, Permutation test) (Fig. 5g), most likely reflecting their degranulation activities during influenza infection.

**Immune dynamics of activated memory influenza-specific CD8+ and CD4+ T cell responses.** To dissect epitope-specific CD8+ and CD4+ T cells during acute IAV infection, we performed patient-specific tetramer staining, utilizing an extensive range of peptide/MHC class-I and class-II tetramers covering the most frequent HLA alleles, with an estimated population coverage of 63-100% across all ethnicities (Fig. 6a, Supplementary Table 4). Following tetramer-associated magnetic enrichment (TAME), robust tetramer+CD8+ T cell populations were detected for all donors ($n = 19$) and specificities tested directly ex vivo (Fig. 6b, c, Supplementary Fig. 6b, c), even when using 10–20% less cell numbers generally required for successful detection of antigen-specific CD8+ T cells in healthy donors[35–37]. In 17 patients, tetramer+CD8+ T cell populations were detected at all the timepoints, while two patients had tetramer+CD8+ T cells directed toward the subdominant B7-NP epitope detectable only at follow-up. For comparison, A2-M1-tetramer+CD8+ T cells were also measured in influenza- patients ($n = 4/4$ detected) (Fig. 6d). Apart from CD8+ T cells, we also identified influenza-specific CD4+ T cells in 5/5 H3N2-infected patients with either HLA-DR*01:01-, DR*04:01- or DR*11:01-HA-tetramers (Fig. 6e, f), despite ~10-fold lower numbers of antigen-specific CD4+ T cells ex vivo. DR*15:02-HA tetramers were also tested but yielded no specific cells in 2/2 patients.

Overall, tetramer precursor frequencies (where $10^{-3}$ denotes 1 per 1000 cells or 1e-3), were stable across timepoints, with no significant differences observed between acute and follow-up (Fig. 6f). Pooled tetramer+CD8+ T cell precursor frequencies in influenza+ patients (mean ± SD, range: 2.62e-4±6.37e-4, 8.07e-7 to 3.56e-3) and influenza- patients (7.00e-5±8.11e-5,

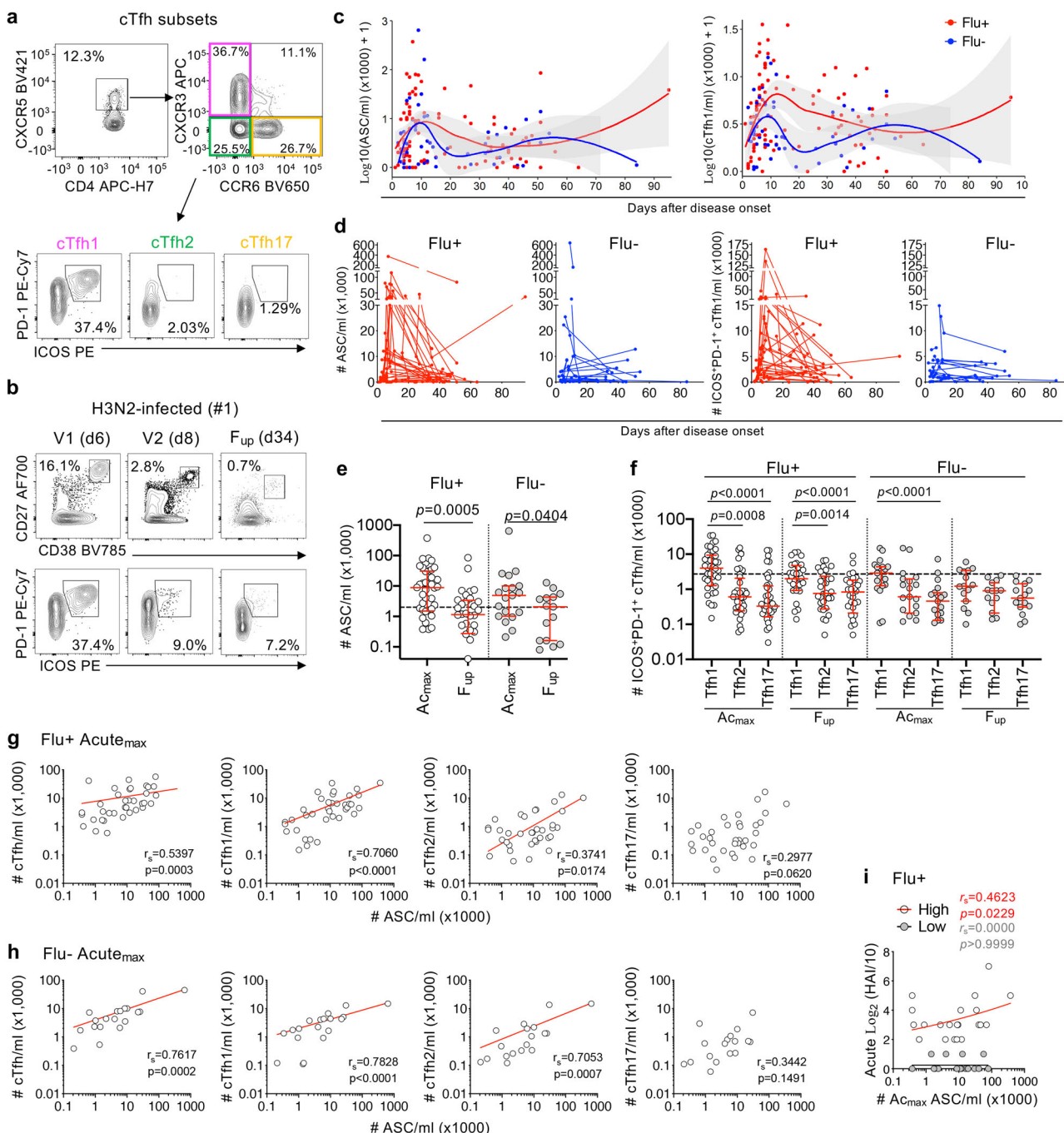

**Fig. 3 Circulating Tfh cells, ASCs and influenza-specific B cell responses. a** Representative FACS plots of CD4+CXCR5+ cTfh subsets (cTfh1/2/17) and expression of activation markers PD-1 and ICOS. **b** Representative FACS plots of CD19+CD20-/loCD27++CD38++ ASCs and PD-1+ICOS+ activated CXCR3+CCR6- cTfh1 cells at acute (V1 and V2) and follow-up ($F_{up}$) timepoints. Days (d) after disease onset are shown in brackets. **c, d** Numbers of ASC and cTfh1 cells of influenza+ (Flu+, red, n=44) and influenza- (Flu-, blue, n=20) patients after disease onset. Gray shading in (**c**) represents 95% confidence intervals. **e** Numbers of the peak ASC responses during acute ILI ($Ac_{max}$) versus $F_{up}$ in influenza+ (open circles, $Ac_{max}$ = 40, $F_{up}$ n = 32) and influenza- patients (gray circles, $Ac_{max}$ n = 19, $F_{up}$ n = 15). **f** Peak cTfh responses of different Tfh1/2/17 subsets at $Ac_{max}$ (influenza+ n = 40, influenza- n = 19) and $F_{up}$ (influenza+ n = 32, influenza- n = 15) timepoints. **g, h** Correlation (two-tailed Spearman correlation coefficient, $r_s$) between $Ac_{max}$ ASC and cTfh responses and (**i**) $Ac_{max}$ ASC responses with HAI antibody titres of high (HAI ≥ 40) and low responders (HAI < 40) during acute infection. **e, f** Red bars indicate the median and IQR for cell numbers. Dashed line is the median baseline level for healthy controls. Statistical significance (0.0001 > p < 0.05) was determined using the two-tailed (**e**) Mann–Whitney test and (**f**) Friedman test. **g–i** Statistically significant p values are shown (0.0001 > p < 0.05).

8.38e-6 to 2.65e-4) were significantly higher 79-fold ($p < 0.0001$, Kruskal–Wallis test) and 21-fold ($p = 0.0026$, Kruskal–Wallis test), respectively, than pooled tetramer+CD4+ T cells (3.30e-6 ±3.20e-6, 6.19e-7 to 1.17e-5) (Supplementary Fig. 6d). Frequencies of pooled tetramer+CD8+ T cells were similar between influenza+ and influenza- patients and fell within the ranges described previously for HLA-A2+ H7N9-influenza+ patients[38,39].

Although the frequency of tetramer+CD8+ T cells were similar between acute and follow-up timepoints in influenza+ patients,

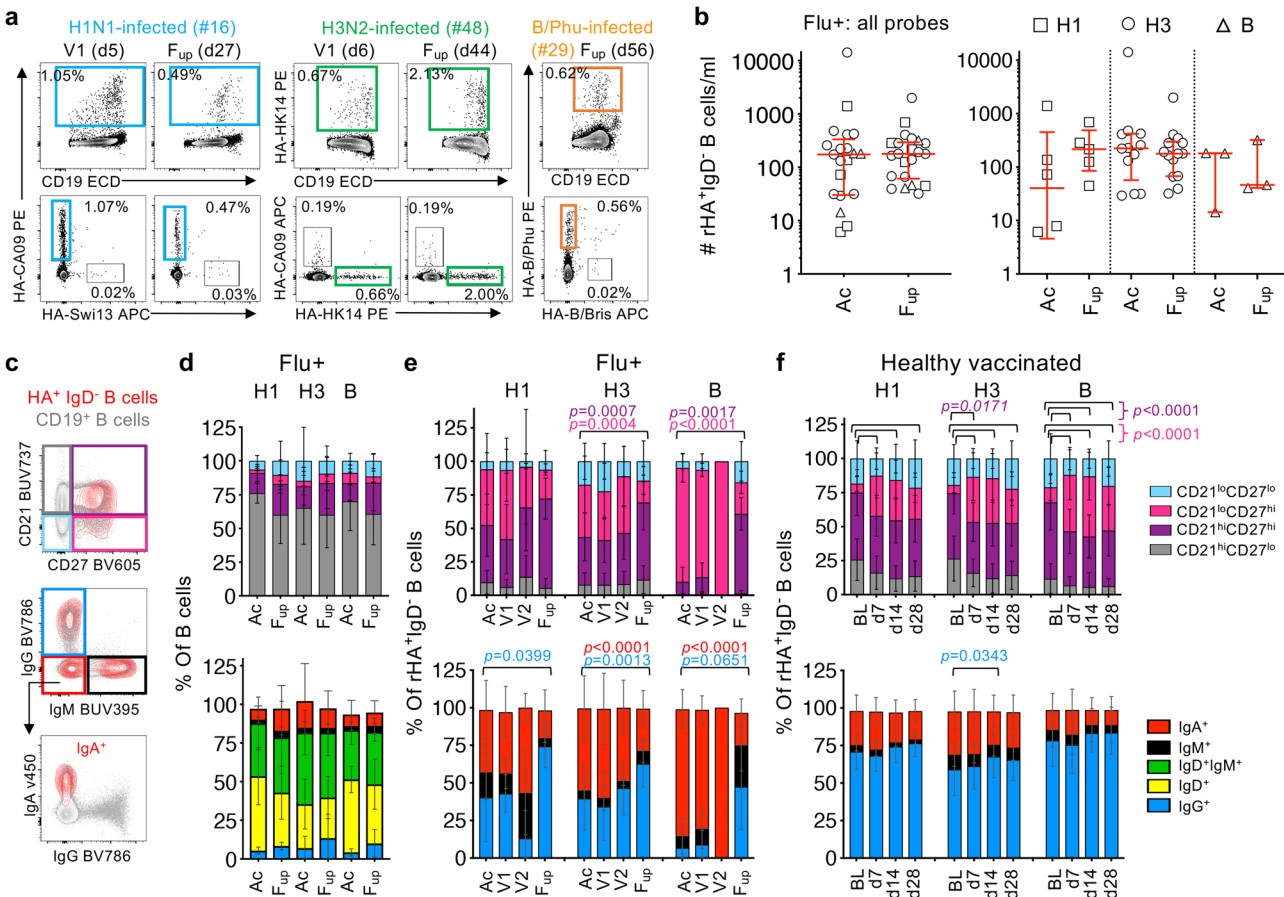

**Fig. 4 Contrasting influenza-specific B cell responses between influenza infection and vaccination. a** Representative FACS plots of H1-, H3- and B-specific rHA+ B cells at V1 and F$_{up}$ timepoints (days after disease onset in brackets). **b** Numbers of total rHA+ B cells and per H1-, H3- and B-specific rHA+ B cells at acute (Ac) (n = 21) and F$_{up}$ timepoints (n = 22). Red bars indicate the median and IQR for cell numbers. **c** Representative overlay FACS plots of live total B cells and rHA+ B cells from IBV-infected patient #29 at F$_{up}$ for phenotype and isotype characterization. **d–f** Phenotype (top panel) and isotype (bottom panel) distributions of (**d**) total B cells and (**e**) rHA+ B cells in influenza+ (Flu+) patients at acute (Ac = V1 and V2, V1, V2) and follow-up timepoints in comparison to (**f**) healthy vaccinated controls (n = 41) pre-vaccination at baseline (BL) and d7, 14 and 28 post-vaccination. **d–f** Bars indicate the mean and SD for frequencies. **e, f** Statistical significance (0.0001 > p < 0.05) was determined using a two-tailed Tukey's multiple comparison test. Colored p values refer to each group legend within the graph.

and influenza+ versus influenza- patients, the activation phenotypes markedly differed. Based on four activation markers expressed (CD38, HLA-DR, PD-1 and CD71), higher levels of activation were detected at the acute timepoint compared to follow-up (Fig. 6g, Supplementary Fig. 6e). Notably, in influenza+ patients, expression of two or more activation markers was most evident between days 6-10 of disease onset for A2-M1+CD8+ T cells, which was in stark contrast to the mainly single-PD-1 expressing or non-activated phenotypes exhibited among the influenza+ follow-up samples, influenza- patients (Fig. 6h), and the parent CD8+ T cell populations. High levels of activation at acute compared to follow-up samples were clearly observed across all tetramer+CD8+ T cell specificities and, despite limited numbers, in tetramer+CD4+ T cells when compared to parental CD8+ and CD4+ T cell populations (Fig. 6i, Supplementary Fig. 7).

Similarly, analysis of CD27/CD45RA/CD95 phenotypes showed significant differences in the activation of tetramer+ CD8+ and CD4+ T cells in influenza+ patients when compared to the parental CD8+ and CD4+ T cell populations at acute and follow-up timepoints. In general, among influenza+ patients, tetramer+CD8+ T cells consisted of central memory-like (Tcm, CD27+CD45RA−) and, to a lesser extent, effector memory-like

(Tem, CD27−CD45RA−) phenotypes, while tetramer+CD4+ T cells were predominantly of Tcm-like cells. (Fig. 6j, Supplementary Fig. 6f). These proportions remained stable across timepoints but were significantly enriched in Tcm compared to the overall CD4+ and CD8+ T cell populations, respectively.

Overall, we show an extensive breadth of highly activated, non-cultured, ex vivo influenza-specific memory CD4+ and CD8+ T cells detected during acute IAV infection, which were still present in a less activated state following recovery.

**Immune correlates of influenza severity.** We also probed clinical, genetic and immune correlates of influenza severity and patients' recovery. We found no significant associations between influenza disease severity (SOFA scores 0–6, clinical presentation: ILI and/or pneumonia), clinical parameters (age, sex, influenza strain, days of disease onset, days in hospital, risk factors and vaccination status) and genetic host factors (IFITM3 SNP alleles). This might be as the majority of hospitalized DISI patients were already in high-risk groups based on their comorbidities/risk factors (86% influenza+, 70% influenza-), including chronic respiratory disease (61% influenza+, 40% influenza-). Furthermore, patients infected with seasonal influenza viruses experienced less severe disease, with

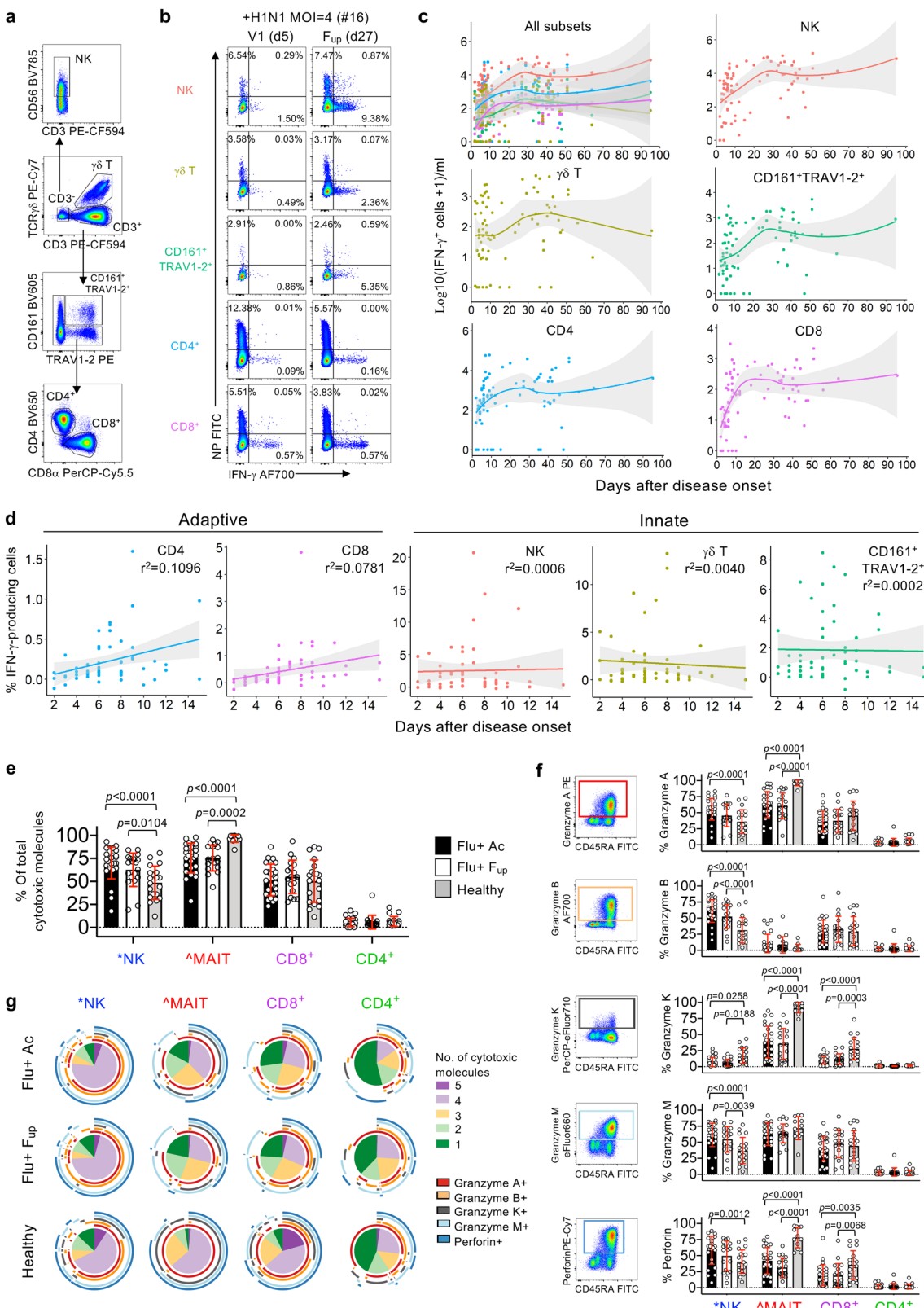

short hospital stays, no requirements for mechanical ventilation, and only one patient requiring ICU for one day, in contrast to the more severe avian H7N9-infected patients[18].

To investigate any potential links between the overall breadth of immune responses in human influenza disease, including cytokines, antibodies, ASCs, activated cTfh1s, IFN-γ-producing innate (NK, γδT, MAIT cells) and adaptive (CD4+ and CD8+

T cells) immune cells, and key clinical/genetic parameters and SOFA scores as a measure for disease severity, we generated heat maps with all of the above mentioned parameters. Unsupervised heat maps of acute and convalescent timepoints were generated for both influenza+ (Fig. 7a, b, Supplementary Data 2 and 3) and influenza− groups, as well as the entire cohort (Supplementary Data 2–6). For influenza+ at the acute timepoints, regions of

**Fig. 5 Innate and adaptive immune responses in seasonal influenza-infected patients. a–d** Data following influenza virus infection assay. **a** Representative FACS plots of innate (NK cells, γδ T cells and CD161+TRAV1-2+ MAIT cells) and adaptive (CD4+ and CD8+ T cells) immune cell subsets gated on live/CD14−/CD19− singlet lymphocytes. **b** Representative FACS plots measuring frequency of infection (intracellular nucleoprotein (NP) staining) and IFN-γ production for each immune cell subset. Infection in PBMCs by intracellular NP-staining showing consistent infectivity rates over time across the donors (Supplementary Fig. 5b). MAIT cells were defined as CD161+TRAV1-2+ and were validated by the MR1-5′OP-RU-tetramer in 51% of samples (Supplementary Fig. 5c,d). **c, d** Numbers and frequencies (n = 37) of influenza-specific IFN-γ-production in patients' immune cell subsets as days of disease onset where 95% confidence intervals are shaded in gray. **e–g** Patient data from PBMCs left over from flow-through fraction following TAME. *NK cells were defined by live/CD14−/CD19−/CD3− cells. ˆMAIT cells were defined by the MR1-5-OP-RU-tetramer and anti-TRAV1-2 antibody. **e** Frequency (mean, SD) of influenza+ (Flu+) patient cells expressing total cytotoxic molecules at acute (includes n = 16 at V1 and n = 11 at subsequent visits i.e., V2, V3, or V4) and follow-up timepoints (n = 19), in comparison to healthy donors (n = 20, except for MAIT subset where n = 12). **f** Representative FACS plots of CD8+ T cells from influenza+ patient and individual frequencies (mean, SD) of granzymes (A, B, K and M) and perforin staining for each cell subset. Dataset n numbers are the same as in (**e**). **g** Pie charts representing the average fractions of cells co-expressing different cytotoxic molecules (slices) and the combinations of granzymes and perforin molecules (arcs). Statistical significance (0.0001 > p < 0.05) was determined using two-tailed (**e**, **f**) Tukey's multiple comparison and (**g**) Permutation tests.

relatively lower cytokine levels of IL-6, MIP-1α, IL-8, MIP-1β, MCP-1, were linked to higher levels of functional IFN-γ-producing immune cells, ASCs and activated cTfh1s (Fig. 7a). As shown in Fig. 1f, these cytokines were strongly positively correlated to each other. At convalescence, lower cytokine regions were linked to higher IFN-γ-producing immune cells, but not ASCs or activated cTfh1 cells, which peaked during acute infection (Fig. 7b). In both acute and convalescent phases, reciprocal regions of higher cytokines levels and lower IFN-γ-producing immune cells (and lower ASCs and activated cTfh1 cells for acute) were observed. However, these observed regions were not defined specifically by SOFA scores or other clinical/genetic parameters.

Based on the functional implications of IFN-γ-producing cells detected here and our previous H7N9 study[12], we analyzed specifically the numbers of IFN-γ-producing cells across acute (V1) and convalescent timepoints (Fup) as a function of patients' disease severity via binned SOFA scores due to unequal numbers of SOFA scores across the cohort, skewing toward low (0–1) SOFA scores, classifying the least severe. At the earliest acute timepoint (V1), decreases in innate IFN-γ-producing γδT and MAIT cells were significantly associated with more severe patients with higher SOFA scores (2–6) (Fig. 7c), but not for NK cells, CD4+ and CD8+ T cells. In contrast, at convalescence, lower adaptive IFN-γ-producing CD4+ and CD8+ T cell responses were associated with higher severity (Fig. 7c). These results indicated that early influenza-specific NK, CD4+ and CD8+ T cell responses were important in driving patients' recovery from influenza disease, and less severe patients had more robust responses at convalescence, while deficiencies in early innate γδT and MAIT cell responses impacted disease severity.

## Discussion
We describe a comprehensive panel of immune cellular networks, in association with clinical and genetic characterization, underlying recovery from influenza infection, which is highly relevant to other infectious diseases. We report that the breadth of robust immune responses to influenza viruses can be measured in peripheral blood prior to patient recovery. Firstly, activated Tfh cells emerged in the blood in parallel with ASCs, both peaking between days 7 and 10 after disease onset in both influenza+ and influenza- groups, prior to patients' recovery, which has not been described before in the blood during and following influenza virus infection. Secondly, hospitalized influenza-patients had increased IL-6, IL-8, MIP-1α/β and IFN-γ inflammatory cytokines, which strongly positively correlated with each other and were associated with lower influenza-specific responses from CD8+ T cells, CD4+ T cells, NK cells, MAIT and γδT cells, and lower ASC and Tfh responses. This is the most extensive study covering innate cells, adaptive cells and 17 cytokines/chemokines.

Thirdly, influenza-specific B cell responses positively correlated with ASC and Tfh responses, but differed to vaccination-induced B-cell responses, revealing contrasting phenotype and isotype characteristics that will inform future B cell-based vaccine designs to promote both IgG+ and IgA+ responses. Forthly, patient-specific peptide/MHC-tetramer staining captured the broadest array of epitope-specific CD8+ and CD4+ T cells during acute IAV infection to date in one study, by utilizing an extensive range (n = 21) of peptide/MHC class-I and class-II tetramers. These tetramers cover the most frequent HLA alleles, with an estimated population coverage of 63–100% across all ethnicities, and provide essential tools for future T cell-immune monitoring of newly-emerging influenza viruses. Lastly, lower influenza-specific γδT and MAIT cell responses were associated with more severe patients with higher SOFA scores, but not for NK cells, CD4+ and CD8+ T cells, showing that early influenza-specific NK, CD4+ and CD8+ T cell responses were important in driving patients' recovery from influenza disease.

Similarly broad and early immunologic analyses of follow-up samples for both influenza+ and influenza- patients after ~30 days post-discharge have not previously been reported. Our work expands on the SHIVERS study, which included follow-up sampling with blood samples collected from mild non-hospitalized and severe hospitalized influenza-infected patients at the acute phase and at 2 weeks post-enrolment during the 2013 New Zealand winter seasons (Aug–Oct)[28]. The SHIVERS cohort was similar to DISI, enrolling 27 hospitalized patients predominantly infected with a H3N2 virus (83%) and 76% with an underlying condition. However, our study differed from the SHIVERS study by the more rapid enrollment following symptom onset (DISI: 4.5 ± 4.4 days (mean ± SD) compared to SHIVERS:10.7 ± 4.8 days) and our patients were hospitalized for longer (DISI median of 4 days (range 1–38 days) compared to SHIVERS median of 2 days (1–10 days)). Rapid recruitment and longer hospitalization with serial sampling enabled the capture of integral immune response kinetics very early during the first week of infection, at a time critical for determining patient outcome.

Our cytokine analyses showed that inflammatory cytokines IL-6, IL-8, MIP-1α, MIP-1β and IFN-γ (an inducer of IP-10) were strongly positively correlated with each other, after adjusting for age and the sampling days from disease onset, and supports previous studies on their collective role as biomarkers for influenza severity[18,20–23]. These cytokine correlations were still robust following recovery at the convalescenct phase, in addition to the emergence of an IFN-α, IL-17 and IL-10 cluster. Furthermore, lower expression of IL-6, IL-8, MIP-1α and MIP-1β were associated with higher IFN-γ-producing immune cells following influenza virus infection assay, as well as ASCs and activated cTfh1 cells, and vice versa.

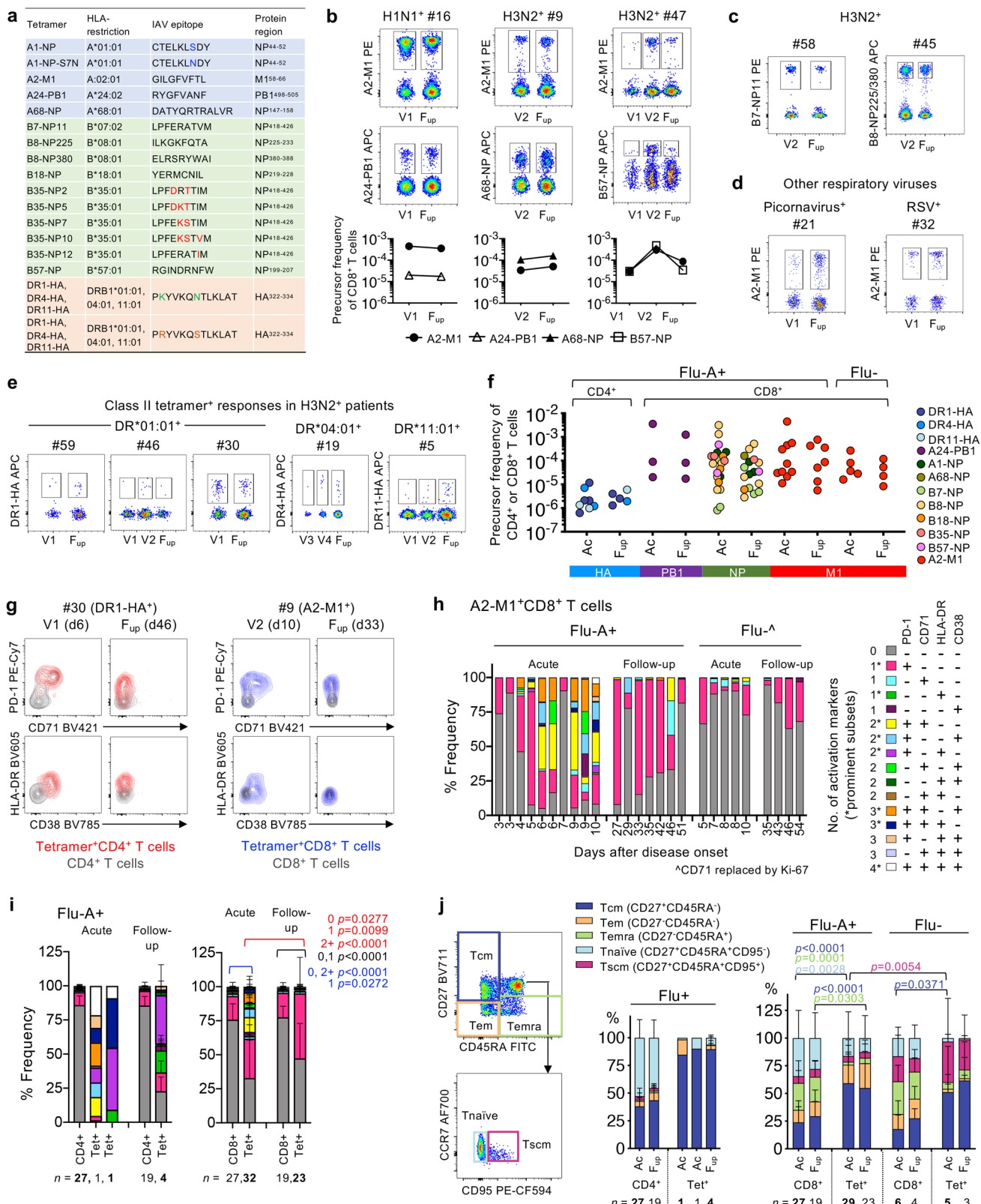

Since genetic host factors, such as specific HLA types and IFITM3 SNPs have been associated with influenza disease severity[15–19] we compared these genetic factors between influenza+ and influenza- patients. However, given that our cohort was mainly white Australian (82% influenza+, 95% influenza-), we found no differences in the number of "universal" or "risk" alleles based on studies describing HLA-A*02:01/03:01 and HLA-B*08:01/18:01/

27:05/57:01 molecules binding universally conserved influenza peptides[40] versus HLA-A*24:02/68:01 types positively correlating with pH1N1 mortality[15] and morbidity[16]. Similarly, for IFITM3 SNP analyses, the non-risk rs12252-T/T homozygous allele was observed in 91% of influenza+ patients (remaining were C/T heterozygous) and 100% of influenza- patients, whereas the risk C/C homozygous allele was absent. The rs34481144 SNP showed a

**Fig. 6 Influenza-specific CD4+ and CD8+ T cell responses. a** List of influenza-specific HLA class I (A, B) and class II (DR) tetramers used in the study. **b**, **c** Concatenated FACS plots of TAME-enriched class I-tetramer+ cells gated on CD8+ T cells from influenza+ patients and (**d**) influenza- patients. Individual tetramer precursor frequencies are shown below for patients in (**b**). **e** Concatenated FACS plots of TAME-enriched class II-tetramer+ cells gated on CD4+ T cells from H3N2-infected patients. **f** Precursor frequencies of tetramer+ cells from influenza+ (Flu+) and influenza- (Flu-) patients at acute (V1, V2, V3 or V4) and follow-up timepoints. **g** Representative overlay FACS plots of activation markers expressed on TAME-enriched tetramer+ cells compared to their unenriched parent population. **h** Frequency of A2-M1+CD8+ T cells from individual influenza A+ (Flu-A+) and influenza- patients expressing different combinations of activation markers PD-1, CD38, HLA-DR and CD71, where CD71 was replaced by Ki-67 in the staining panel for influenza- patients. **i** Overall activation status of TAME-enriched tetramer+ cells compared to their unenriched parent population of CD4+ or CD8+ T cells in influenza+ patients. **j** T cell differentiation phenotype of TAME-enriched tetramer+ cells in relation to the unenriched parent population of CD4+ or CD8+ T cells. **i, j** Mean and SD are shown for all acute and follow-up timepoints, except for the acute tetramer+CD4+ group ($n = 2$), which were plotted individually. Statistical significance ($0.0001 > p < 0.05$) was determined using two-tailed Tukey's multiple comparison test for (**i**) number of activation markers present (0, 1 or 2+) and (**j**) T cell differentiation subsets.

higher frequency of the A/G-heterozygous allele for both groups (41% and 50%, respectively), with equal or similar proportions of the risk A/A and non-risk G/G homozygous alleles. Further analyses of influenza+ patients with the risk A/A rs34481144 SNP versus influenza+ patients with A/G-heterozygous allele and non-risk G/G homozygous alleles showed no differences in disease severity by SOFA scores (Supplementary Fig. 8).

For antibody responses, HAI antibody titres were lower in influenza+ patients during acute infection compared to influenza- patients and healthy controls, but increased significantly following recovery against the infected strain and previous strains from the same subtype. It is unclear whether the SHIVERS study compared antibody responses between acute versus convalescent phase within mild non-hospitalized and severe hospitalized patient groups, although they did not observe any differences when comparing antibody responses of mild versus severe groups at both acute and convalescent timepoints[28]. Given that we observed back-boosting of cross-reactive antibody responses post-infection at the follow-up timepoint, we postulate that having a natural influenza virus infection, compared to a vaccine-induced response, elicits more effective cross-reactive antibody responses post viral clearance warranting further investigations.

In a study to measure influenza-specific B and T cell responses following seasonal IIV in healthy individuals[4], vaccine-induced antibody responses positively correlated with increases in ASCs, influenza-specific memory B cells and a subset of activated cTfh cells. Similarly, during acute influenza disease, we show prominent activation of cTfh1 cells, which correlated with ASC responses and memory influenza-specific B cells. In contrast, numbers of cTfh1 did not correlate with antibody titres at acute, follow-up or fold-change titres. Only ASC numbers positively correlated with higher acute antibody titres of 40 and above, again supporting a cut-off titre of 40 as a measure of protection. A very small subset of CXCR5+PD-1high cTfh cells in the blood have been shown to correspond to CXCR5+PD-1high germinal centre Tfh cells by sharing transcriptional and TCR repertoire profiles[41,42]. We analyzed CXCR5+PD-1high expression on total CD4+ T cells in a small subset of influenza+ patients and also found very low levels in the blood at both acute and follow-up timepoints (0-0.2%, Supplementary Fig. 9). This is in contrast to a more prominent population found in lymph nodes[41] and tonsils[42]. The role of cTfh cells as surrogates to germinal center Tfh cells have been described during dengue infection[43] and very recently by our group in a COVID-19 patient[44]. However, to the best of our knowledge, there have been no previous reports on human cTfh cells following natural influenza virus infection.

Interestingly, phenotypic analysis of the memory influenza-specific B cell response showed contrasting profiles induced by infection versus vaccination. During acute infection, influenza-specific B cells were of an activated-memory phenotype comprising IgG and IgA isotypes but were predominantly a resting-memory IgG+ population at convalescence. Conversely, IIV induced an expansion of activated-memory influenza-specific B cells but there were no changes in the isotype distribution which were predominantly IgG+ B cells, given that the influenza vaccine is not delivered to a mucosal site. IgA+ memory B cells are able to localize in the blood and mucosal sites of inflammation such as the respiratory tract during influenza virus infection to provide a rapid immune response, while IgG+ memory B cells generally circulate throughout the body[45]. Therefore, in accordance with our earlier antibody findings, we suggest that future B cell-based influenza vaccines promoting cross-reactive B cell responses encompassing both activated-memory IgA+ and IgG+ subclasses may provide further protection at the site of infection.

Our study utilized a whole live virus assay to measure the kinetics of influenza-specific responses in NK cells, γδ T cells and CD161+TRAV1-2+ MAIT cells (innate-like), and CD8+ and CD4+ T cells (adaptive). We observed robust and increasing proportions of adaptive CD8+ and CD4+ T cell responses within the first 2 weeks of infection, despite the patients' overall lymphphopenic state, illustrating the ability of these cell types to drive patient recovery. In contrast, innate lymphocytes did not increase in their ability to respond to influenza virus infection by cytokine production, and lower IFN-γ-producing γδ T cells and MAIT cells were significantly associated with higher disease severity based on SOFA scores at the earliest V1 timepoint. Similarly, an H7N9 study showed the robustness of the adaptive CD8+ and CD4+ T cell-immune response in driving recovery from severe H7N9 disease[12]. These findings have important implications for future influenza vaccines because, apart from the CD4+ T cell help compartment, IIV fails to induce CD8+ T cell responses nor any innate responses from NK cells, γδ T cells and MAIT cells, as previously reported[4].

We probed influenza-specific T cells ex vivo using a range of peptide/MHC class I and class II tetramers covering the most frequent HLA alleles with an estimated population coverage of 63–100% across all ethnicities. Tetramer+ CD8+ and CD4+ T cells were highly activated at acute timepoints, compared to a less activated profile at follow-up, and mainly expressed combinations of PD-1 with CD38 and CD71, and HLA-DR to a lesser extent. In contrast to severe H7N9 influenza[13], Ebola[46] and COVID-19 disease[44], with prominent CD38+HLA-DR+ populations in the total CD8/CD4 compartments, we did not observe marked populations of CD38+HLA-DR+ T cells in either tetramer+ or parent populations, perhaps reflecting a less severe cohort of diseased patients in our DISI study. Nonetheless, we provide acute and convalescent populations of rare HA306-318-specific CD4+ T cells in H3N2-infected patients that were HLA-DR*01:01+ ($n = 3$), HLA-DR*04:01+ ($n = 1$) or HLA-DR*11:01+ ($n = 1$), which complements other studies assessing cells in HLA-DR*04:01+ healthy

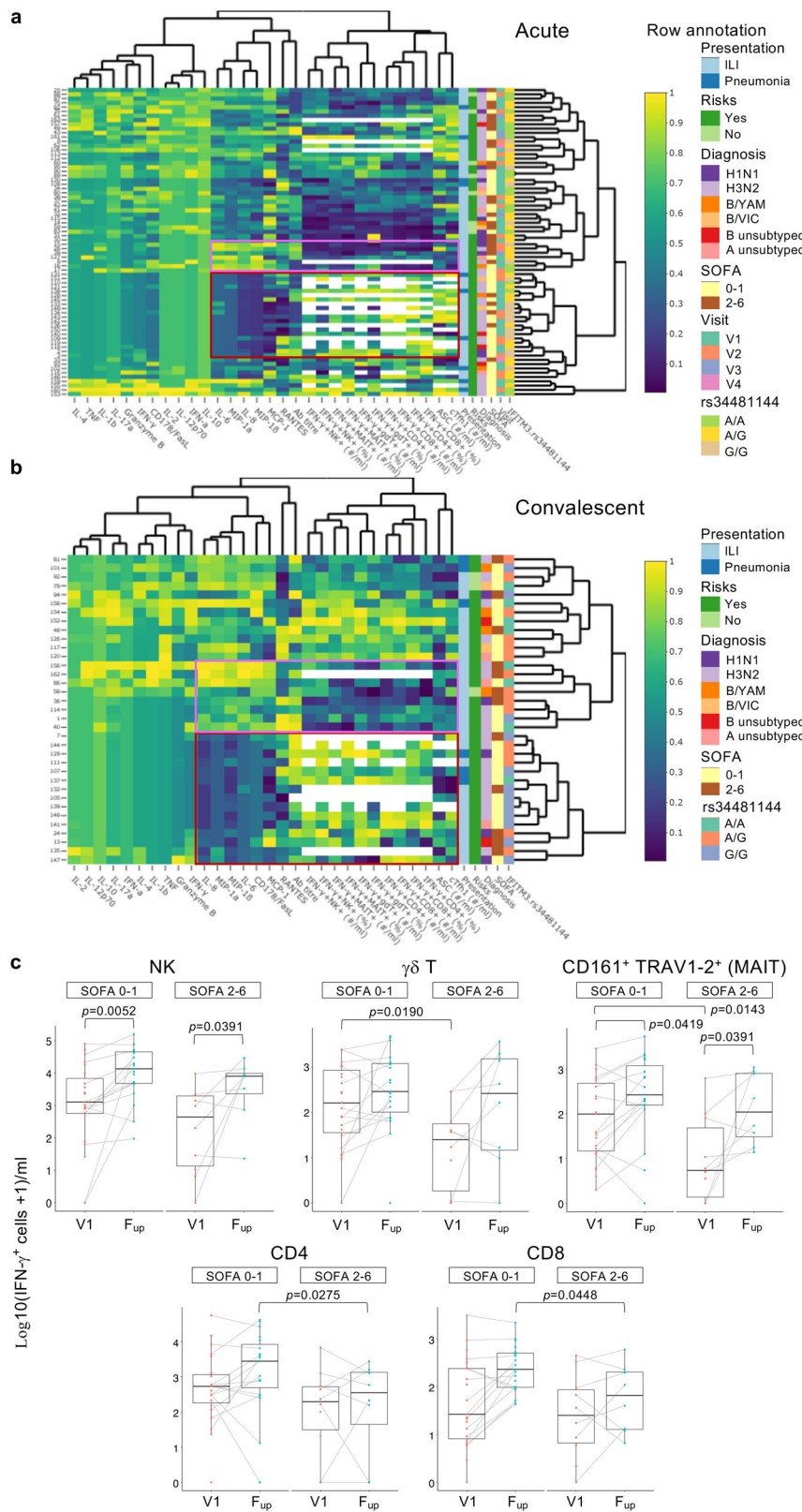

donors and rheumatoid arthritis patients following influenza vaccination[47,48]. Although we may have missed strong responses to rarer epitopes and HLA presenting molecules, our results represent the broadest array of influenza-specific tetramer+ T cell responses during acute influenza infection, providing essential tools for future T cell-immune monitoring of newly-emerging influenza viruses.

Intriguingly, influenza-specific T cells were rarely above 1% of total CD8+ T cells in the blood and were not significantly increased during acute infection compared to the convalescence phase. This is in stark contrast to blood and lymphoid infections such as EBV and HIV-1 where blood levels can be very high (>10% for a single specificity). It is tempting to speculate that perhaps they

**Fig. 7 Analyses of immune responses and clinical and genetic host factors. a,b** Unsupervised heatmaps of immune, clinical and genetic parameters in influenza+ patients at (**a**), acute and (**b**), convalescent timepoints. Scale visualizes each variable using the Empirical Percentile Transformation. A value of 1 means that the measure was higher than 100% of the other samples for that measure. Each variable was transformed independently. Interactive heatmaps are shown in Supplementary Data 2–6 for influenza+, influenza- and combined datasets. Regions of low (maroon) and high (pink) cytokine clusters are boxed. **c** Box plots of IFN-γ-producing cells following influenza virus infection assay ($n = 37$), at the earliest acute (V1) and convalescent ($F_{up}$) timepoints as a function of patients' disease severity via binned SOFA scores of 0–1 versus 2–6. Box plots represent the median (middle bar), 75% quantile (upper hinge), and 25% quantile (lower hinge), with whiskers extending 1.5 times the inter-quartile range. Nonparametric Wilcoxon rank sum test with continuity correction was used for comparisons between SOFA categories (two-tailed). Wilcoxon signed rank test (a paired test, two-tailed) was used to compare differences between V1 and $F_{up}$ among the same individuals. Tests were carried out on the actual data, although plots were on a log10+1 scale for ease of visualization. Statistically significant $p$ values are shown ($0.0001 > p < 0.05$).

may be migrating to the respiratory tract during acute influenza infection, although we do not have BAL samples from influenza+ patients to test this hypothesis. However, human lung cells from deceased organ donors labeled with influenza class I tetramers (HLA-A2-M1$_{58}$ and HLA-B57-NP$_{199}$) showed high frequencies of up to 8% of influenza-specific CD8$^+$ T cells within the memory CD45RO$^+$CD8$^+$ T cell population[49]. We speculate that perhaps these cell populations in the lung may expand to even greater numbers during acute infection. In support, an RSV infection study showed higher frequencies of RSV-specific CD8$^+$ T cells in the BAL compared to the blood at day 10 and day 28 post-infection[50]. Blood RSV-specific CD8$^+$ T cells peaked on day 10 (<0.5% of CD8$^+$ T cells) and then contracted by day 28, whereas BAL RSV-specific CD8$^+$ T cells continued to increase from day 10 (~1–2%) to day 28 (~10%) in the convalescence phase.

There are limitations to this study. The DISI cohort included influenza+ patients who were ill enough to be admitted to hospital, thus representing a skewed population of severe influenza cases. The majority of these patients were also already at high-risk, with 86% having pre-existing comorbidities. Therefore, some findings of our study may not be applicable to the general population that experience mild influenza symptoms, have no comorbidities or are very young healthy individuals. We acknowledge that our DISI study included a small cohort ($n = 65$: $n = 45$ influenza+ and $n = 20$ influenza-), but we performed a comprehensive study, which included longitudinal sampling during the hospital stay and a follow-up sample at ~30 days after discharge. Although we cannot readily distinguish cause from effect, different hypotheses on the immune mechanisms underlying influenza disease severity remain. However, our DISI study provides a valuable and important body of work that provides key insights into understanding immune responses during severe influenza disease.

Tracking antibody, B and T cell responses (and other immune mediators) may be predictive of patients' severity or recovery from influenza virus infections that require hospitalization. Our current analyses of immunological and virological parameters within a clinical hospital framework provide a unique and key dataset on the mechanisms underpinning influenza severity and susceptibility in the human population. Furthermore, these efforts represent a substantial progression to elucidate the host immune responses underlying the recovery from this acute disease as well as other infectious diseases such as COVID-19[44].

## Methods

**Study participants and design**. The DISI cohort enrolled consenting adult patients admitted to The Alfred hospital during the 2014–2017 peak influenza seasons with ILI. We recruited 44 influenza-positive patients and 20 influenza-negative patients with other respiratory diseases. Bloods were collected within 24–72 h of hospital admission (Visit 1, V1), every 2–5 days until discharge (V2, V3, V4 etc), then followed up ~30 days later. Nasal swabs were collected at V1 and V2 timepoints while in hospital. Clinical data collection included vaccination status,

sequential SOFA score[29] as a measure of disease severity, and any significant risk factors (Supplementary Table 1, Supplementary Data 1).

The H7N9-infected hospitalized patient cohort ($n = 18$) has previously been described[18]. Healthy adults vaccinated in 2015 (TIV, $n = 16$) and 2016 (QIV, $n = 26$) have previously been described in detail[4], where blood samples were collected prior to vaccination (day -1 or 0) and on days 7, 14 and 28 following vaccination. Healthy blood donors were recruited from The University of Melbourne and Deepdene Surgery. Buffy packs were sourced from Australian Red Cross Lifeblood, respectively (Supplementary Table 3).

Human experimental work was conducted according to the Declaration of Helsinki Principles and according to the Australian National Health and Medical Research Council Code of Practice. All participants provided written informed consent prior to the study. The study was approved by the Alfred Hospital (ID #280/14) and University of Melbourne (ID #1442952.1 and #1443389.4) Human Research Ethics Committees. Up to 40 mls of DISI patient blood was collected in sodium heparin tubes (including 1 serum tube) and processed within 24 h. Up to 60 mls was collected from healthy donors and ~50–60 ml for buffy packs. PBMCs were isolated by Ficoll-Paque (GE Healthcare, Uppsala, Sweden) density-gradient centrifugation and cryopreserved. DNA was extracted from the granulocyte layer using a QIAamp DNA Mini Kit (Qiagen, Hilden, Germany, #51306) and sent to the Victorian Transplantation and Immunogenetics Service (Australian Red Cross Lifeblood, West Melbourne, Victoria, Australia) for HLA class I and class II molecular typing using the Luminex platform and microsphere technology (One Lambda, Canoga Park, CA, USA), with LABType SSO HLA typing kits (One Lambda, #RSSO1E47, #RSSO2345). Nasal swab samples were kept cold during transport and stored at −80 °C within 4 h.

In cases where fewer than the total number of donors were analyzed (Supplementary Data 7), DISI patients were selected for influenza-specific cellular assays based on their influenza status and remaining sample availability, such as the class I and II TAME experiments which were carried out on IAV$^+$ patients and relied on HLA type and tetramer availability. HIV-positive patients ($n = 5$, Supplementary Data 1) were not included in such experiments to fulfil biocontainment regulations. No other blinding or randomization protocols were applied and no outliers (i.e., the two death patients) were excluded.

**IFITM3 SNP analysis**. Amplification and sequencing of *exon 1* rs12252 region were performed by PCR on genomic DNA using IFITM3 forward and reverse primers (Supplementary Table 5) with *Taq* DNA polymerase (Qiagen, #201209)[51]. Amplification of the rs34481144 promotor region was performed using the same forward and reverse primers but with the Phusion High-Fidelity PCR Master Mix with HF Buffer (New England Biolabs, #M0531S), and were sequenced with a different forward and reverse primer[17]. All primers are listed in Supplementary Table 5.

**Cytokine analysis**. Patient sera or plasma was diluted 1:4 for performing cytokine bead assay (CBA) using the Human CBA Kit (BD Biosciences, San Jose, California, USA, #558265), according to manufacturer's instructions. Capture beads included IL-2 (#558270), IL-4 (#558272), IL-6 (#558276), IL-8 (#558277), IL-10 (#558274), IL-12p70 (#558283), IL-17A (#560383), IL-1β (#558279), IFN-α (#560379), MIP-1α (#558325), MIP-1β (#558288), MCP-1 (#558287), CD178/FasL (#558330), granzyme B (#560304), RANTES (#558324), TNF (#558273) and IFN-γ (#558269). Samples were also diluted 1:100 for RANTES. Samples were acquired on a BD CantoII capturing all events. A limited panel of cytokines for H7N9 patients have already been described[18].

**Viral sequencing**. The IAV HA were sequenced by NGS using the method published before[52]. Briefly, virus RNA was extracted from nasal swabs or cultured virus using QIAamp Viral RNA Mini Kit (Qiagen, #52906). Influenza A or B multiplex RT-PCR reactions were set up using 3 μl of RNA and 3 μl of primer cocktail (Supplementary Table 5) with SuperScript III one-step RT-PCR system with Platinum Taq high-fidelity DNA polymerase (Invitrogen, #12574030). PCR products were further fragmented to 200 bp with Ion Xpress Plus fragment library kit (Life

Technologies, #4471269) and then ligated to Ion Xpress barcode adapters 1–96 (Life Technologies, #4474517), then samples with different barcodes were pooled. AMPure XP reagent (Agencourt, #A63881) was used to clean up the library pool and quantified with the Ion library quantitation kit (Life Technologies, #4468802), 10 pM of final library concentration was used to prepare template-positive Ion Sphere particles on the Ion OneTouch 2 instrument (Life Technologies), Ion 316 Chip version 2 (Life Technologies, #4488149) was used for sequencing on the Ion Torrent PGM. FluLINE pipeline was used for data analysis to produce consensus sequence[52]. HA phylogenetic analysis with neighbor-joining trees were done using the Geneious 9.0.4 software (Biomatters Ltd).

**Viruses**. Influenza A (A/H1N1/California/7/2009, A/H3N2/Switzerland/9715293/ 2013 and H3N2/Hong Kong/4801/2014) and B (B/YAM/Phuket/3073/2013 and B/ VIC/Brisbane/60/2008) viruses were grown in 10-day old embryonated chicken eggs at 33 °C for 72 h. Allantoic fluid was harvested and titrated using standard plaque assays in MDCK cells for use in the influenza infection assays. Other viruses (Supplementary Table 2) were sourced from the WHO Collaborating Centre for Reference and Research on Influenza (Melbourne, Australia).

**Hemagglutination inhibition assay**. RDE-treated sera or plasma samples were assessed for antibody titres against a panel of influenza viruses (Supplementary Table 2) using standard HAI assays with 1% turkey red blood cells (H1N1 and IBV) or 1% guinea pig red blood cells in the presence of oseltamivir (H3N2), according to WHO guidelines. HAI titres were reported as the reciprocal of the highest dilution of serum where hemagglutination was completely inhibited, then divided by 10 and $\log_2$-transformed.

**Flow cytometry**. Fresh whole blood was used to measure absolute cell numbers using BD TruCount tubes and BD MultiTest (CD3 FITC/CD16+CD56 PE/CD45 PerCP/CD19 APC, #340500) according to manufacturer's instructions (BD Biosciences), as well as Tfh and ASC populations as described[4]. Thawed PBMCs from influenza+ patients were stained with recombinant (r) HA probes from H1 (California/7/2009), H3 (Switzerland/9715293/2013 before 2016 or Hong Kong/ 4801/2014 since 2016) and B strains (Phuket/3073/2013 or Brisbane/60/2008)[4,33], essentially as described, except anti-IgA was not included in rH1 and rH3 probe panels. PBMCs were stained with rHA probes conjugated to SA-PE and SA-APC and antibody cocktail[4,33] in 150 ul of MACS buffer for 30 min on ice, washed and fixed with 1% PFA for 30 min before acquisition. PBMCs were cell surface stained in 50 ul with panels 1 and 3 (Supplementary Data 8) for 30 min on ice, fixed then permeabilized and stained with intracellular antibodies from panels 1 and 3 using the BD Cytofix/Cytoperm Kit (#554714) according to manufacturer's instructions. PBMCs were cell surface stained with panel 2 (Supplementary Data 8) following TAME as detailed below. TruCount/MultiTest samples were acquired on a BD CantoII to achieve 3000 bead events (~30,000 lymphocytes). All other samples were acquired on a BD LSRII Fortessa collecting all events. Flow cytometry data were analyzed using FlowJo v10 software.

**Influenza virus infection assay**. Thawed PBMCs (<1.5e6) from DISI influenza+ patients were cultured overnight at 37 °C/5% CO$_2$ with their cognate live egg-grown virus (MOI = 4), essentially as described[4,12], for a total of 22 h before cells were harvested, then cell surface and intracellularly stained with Panel 1a or 1b (Supplementary Data 8) using the BD Cytofix/Cytoperm Kit (#554714) according to manufacturer's instructions. A no virus culture was included as a background staining control, which was subtracted from the virus culture. All events were collected on a BD LSRII Fortessa.

**Tetramer-associated magnetic enrichment**. TAME was performed on thawed PBMCs (3-38e6) for the detection of influenza-specific CD4$^+$ and CD8$^+$ T cells, as described[37,38]. Peptide/MHC class I and class II monomers (Fig. 6a) were generated in-house[53,54], before 8:1 molar ratio conjugation with either PE-streptavidin (SA) or APC-SA (BD Biosciences) to form tetramers. Briefly, cells were stained (1:100-200 dilution) with PE- (A2-M1, B7-NP, B8-NPs, B18-NP and B35-NPs) or APC-labelled (A1-NPs, A2-M1, A24-PB1, A68-NP, B57-NP and DR1/4/11-HAs) tetramers for 1 h at room temperature before dual-magnetic enrichment with anti-PE and anti-APC MicroBeads with LS columns (Miltenyi Biotec, Bergisch Gald-bach Germany) as described[37,39]. Tetramers restricted to the same HLA were added together at the same dilution. Unenriched, enriched and flow-through fractions were cell surface stained with Panel 2a (influenza+) or 2b (influenza-, follow-up samples only) (Supplementary Data 8) for 30 min on ice, fixed with 1% PFA, then acquired by flow cytometry. Panel 2b samples were fixed and permeabilized using the eBioscience™ Foxp3/Transcription Factor Staining Buffer Set (Thermo Fisher Scientific, Carlsbad, CA, USA, #00-5523-00). Samples with cell events just below 10, especially for tetramer+CD4$^+$ T cells, were not further characterized phenotypically using these staining panels. All events were collected on a BD LSRII Fortessa.

**Expression of cytolytic molecules**. Flow-through fractions of DISI patients following TAME or thawed PBMCs from healthy donors (Supplementary Table 3)

were stained with Panel 3 (Supplementary Data 8) for measuring granzymes (A, B, K and M) and perforin using the Foxp3/Transcription Factor Staining Buffer Set. All events were collected on a BD LSRII Fortessa.

**Statistical analyses**. Statistical significance was assessed using GraphPad Prism v8 software unless stated otherwise. Mann–Whitney (unpaired) and Wilcoxin (paired) tests were two-tailed. Friedman (matched) and Kruskal–Wallis (unmatched) tests were used to compare more than two groups. Tukey's multiple comparison test compared row means between more than two groups. Partial correlation plots (Fig. 1f) showing significant (FDR-adjusted $p$ value < 0.05) colored squares were partialed to account for the variance caused by $\log_{10}$(age) and the sampling days after disease onset. To build antibody landscapes (Fig. 2f), local regression (LOESS) was applied in R 3.5.3[55]. LOESS with 95% confidence intervals was also used to plot Figs. 3c, 5c and 5d. Linear relationships between day of sample and percent of IFN-γ producing cells were assessed with ordinary least squares (lm function) in R with 95% confidence intervals shaded in gray and correlation coefficients ($r^2$) reported in Fig. 5d. Correlations (Fig. 3g-i, Supplementary Fig. 4) were assessed using Spearman's correlation coefficient ($r_s$). Polyfunctionality pie charts (Fig. 5g) were generated using Pestle v1.8 and Spice v5 software[56] and $p$ values were calculated using Permutation Test. $P$ values lower than 0.05 were considered statistically significant and exact $p$ values are shown in the figure. All statistical tests are indicated in the figure legends.

**Reporting summary**. Further information on research design is available in the Nature Research Reporting Summary linked to this article.

## Data availability

Source data are provided with this paper. All other data are provided in the article and its Supplementary files or from the corresponding author upon reasonable request. The viral sequences isolated from nasal swabs that support the findings of this study are available in the GenBank database with accession numbers MW674875, MW674876, MW674877, MW674878 and MW674796. Source data are provided with this paper.

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

## Acknowledgements

We thank Jill Garlick, Janine Roney, and the research nurses at the Alfred Hospital, Nicola Bird, Kim Harland, Louise Carolan and Bernadette McCudden for their technical assistance. Thank you to Professors David Fairlie and Jim McCluskey for the MR1-5'OP-RU tetramer. We also thank the study participants for providing blood for research purposes. This work was supported by The Australian National Health and Medical Research Council (NHMRC) NHMRC Program Grant (1071916) to K.K., S.J.T., N.L.G., A.K., D.C.J., L.E.B., W.C., and a NIAID UO1 grant 1U01AI144616-01 "Dissection of Influenza Vaccination and Infection for Childhood Immunity" (DIVINCI) to K.K. and P.G.T. THON is supported by the NHMRC EL1 Fellowship (#1194036). M.K., M.Au, L.H. and S.N. were recipients of Melbourne International Research Scholarship and Melbourne International Fee Remission Scholarship. C.E.S. has received funding from the European Union's Horizon 2020 research and innovation program under the Marie Skłodowska-Curie grant agreement No. 792532. E.B.C. is a NHMRC Peter Doherty Fellow. S.S. was a recipient Victoria India Doctoral Scholarship and Melbourne International Fee Remission Scholarship, University of Melbourne. The Melbourne WHO Collaborating Centre for Reference Research on Influenza is supported by the Australian Government Department of Health. S.G. is a NHMRC SRF-Level A Fellow. J.R. is supported by an ARC Laureate fellowship. N.L.G. is supported by a NHMRC Ideas grant, ARC Discovery Project and ARC Future Fellowship. P.G.T. is supported by the St. Jude Center of Excellence for Influenza Research and Surveillance (NIAID Contract HHSN27220140006C), R01 AI 107625, R01AI136514 and ALSAC. ACC is a NHMRC Career Development (level 2) Fellow. K.K. was supported by a NHMRC Senior Research Fellowship Level B (#1102792), NHMRC Investigator Grant (#1173871) and the University of Melbourne Dame Kate Campbell Fellowship.

## Author contributions

K.K. supervised and lead the study. K.K., T.H.O.N., M.K., C.E.S. and L.L. designed the experiments. T.H.O.N., M.K., C.E.S., L.L., L.G., S.S., E.B.C., M.Au, L.H., Z.W., S.N., A.F., X.X., M.Ab, K.L.L. and Y.D. performed experiments. T.H.O.N., M.K., C.E.S., J.C.C., L.L., L.G., E.K.A., T.B., E.B.C., M.Au, L.H., P.G., A.C.H., P.G.T. and K.K. analyzed data. A.K.W. and S.J.K. provided invaluable rHA probes. S.G. and J.R. provided invaluable pMHC-I/II tetramers. J.C., J.X., T.C.K. and A.C.C. recruited the patient cohorts. K.K., A.C.C., T.C.K., S.J.T., P.C.D., D.J., L.E.B., N.L.G. and W.C. provided intellectual input into the study design and data interpretation. T.H.O.N., P.G.T., A.C.C. and K.K. wrote the paper. All authors reviewed and approved the paper.

## Competing interests

The authors declare no competing interests.

## Additional information

[1]Department of Microbiology and Immunology, University of Melbourne, at the Peter Doherty Institute for Infection and Immunity, Parkville, VIC, Australia. [2]Department of Hematopoiesis, Sanquin Research and Landsteiner Laboratory, Amsterdam UMC, University of Amsterdam, Amsterdam, Netherlands. [3]Department of Immunology, St Jude Children's Research Hospital, Memphis, TN, USA. [4]Biology Department, École Normale Supérieure Paris-Saclay, Université Paris-Saclay Cachan, Cachan, France. [5]Melbourne Sexual Health Centre and Department of Infectious Diseases, Alfred Hospital and Central Clinical School, Monash University, Melbourne, VIC, Australia. [6]ARC Centre for Excellence in Convergent Bio-Nano Science and Technology, University of Melbourne, Parkville, VIC, Australia. [7]World Health Organisation (WHO) Collaborating Centre for Reference and Research on Influenza, at The Peter Doherty Institute for Infection and Immunity, Melbourne, VIC, Australia. [8]Infection and Immunity Program & Department of Biochemistry and Molecular Biology, Biomedicine Discovery Institute, Monash University, Clayton, VIC, Australia. [9]Australian Research Council Centre of Excellence for Advanced Molecular Imaging, Monash University, Clayton, VIC, Australia. [10]Department of Biochemistry and Genetics, La Trobe Institute For Molecular Science, La Trobe University, Bundoora, VIC, Australia. [11]Institute of Infection and Immunity, Cardiff University School of Medicine, Heath Park, Cardiff, UK. [12]Deepdene Surgery, Deepdene, VIC, Australia. [13]Shanghai Public Health Clinical Centre and Institutes of Biomedical Sciences, Key Laboratory of Medical Molecular Virology of Ministry of Education/Health, Shanghai Medical College, Fudan University, Shanghai, China. [14]Department of Microbiology, Biomedicine Discovery Institute, Monash University, Clayton, VIC, Australia. [15]Department of Allergy, Immunology and Respiratory Medicine, The Alfred Hospital, Melbourne, VIC, Australia. [16]Department of Medicine, Monash University, Central Clinical School, The Alfred Hospital, Melbourne, VIC, Australia. [17]School of Public Health and Preventive Medicine, Monash University, Melbourne, VIC, Australia. [18]Infection Prevention and Healthcare Epidemiology Unit, Alfred Health, Melbourne, VIC, Australia. [19]These authors jointly supervised this work: Kedzierska K, Cheng AC. ✉email: allen.cheng@monash.edu; kkedz@unimelb.edu.au

