## [Peer Review File · Nature Communications]

Editorial Note: The figure on page 12 in this Peer Review File has been amended to remove third-party material where no permission to publish could be obtained.

REVIEWER COMMENTS

Reviewer #1 (Remarks to the Author):

I commend the authors on their efforts here to explore the immunopathogenesis of influenza infection and find this manuscript very interesting and useful. Comments:

1. I would like to see some discussion of the weaknesses of the study. In particular the authors should discuss the fact that they are only observing FLU+ patients who were sick enough to present for testing/care. This is a skewed population and represents the tip of the iceberg in terms of influenza cases worldwide. Some of their finding may not be applicable to the less ill individuals or individuals at less risk of complication/ more severe illness such as very young healthy individuals. Also there should be a discussion on how this study was done overall on a very small number of people and it is unclear to me how diverse these individuals were in terms of age, sex, race, etc.
2. It would be useful to have some more information about the patients they enrolled in terms of demographics, etc.
3. The statement that their work supports The Who recommendations for vaccination should be changed or removed. They looked at a population of individuals who presented for care and the ones who were flu positive were less likely to be vaccinated, but the flu negatives were also presenting for care. There are multiple confounding factors here. First is that there is no way to know for sure that the FLU negative did not have influenza, but the window for detection was missed. The skewed population they looked at may not represent the general population at large, i.e. healthy individuals, children, and so on, and so one must be careful here. They should either remove it or simply state that in this particular population of people presenting to hospital it suggests that vaccination may be beneficial.
4. Lastly I think it would be helpful if the authors could better summarize their conclusions from all of their work as they end the discussion to highlight the key points of their findings. As it is I feel quite a bit of the information gets lost in the discussion and may cause the findings to not be appreciated for what they are.

Reviewer #2 (Remarks to the Author):

This is wonderfully rich data set, but it could be presented in a way that permits more patterns exhibited by individual subjects to be apparent. Most data are presented in aggregate and although the last Figure analyzes them on a subject-to-subject basis, it will be difficult for the reader to extract specific features of the immune response to infection that they might be interested in. It is recommended that the paper focus on key issues that can be presented in a more centered conceptual framework, allowing more space for discussion and re-analyses of key data and place the ancillary issues and data into the supplemental data, if they desire. For example, the antibody data and lineage relationships in Figure 2 do not appear to be essential to the main issues addressed in this study, nor do the SNP data in Figure 1.

Recommendations

1. The choice of the major control group seems problematic. Most are pathogen-infected and several are diagnosed with RSV or parainfluenza virus. These infections would have unique and not insignificant distinctive features of induced immunity. The logic for this is not clear and would seem to derive more out of ease of sampling subjects in parallel. Data for most parameters should be presented for a similarly sized cohort of healthy controls subjects that are age and comorbidities-matched. The changes in cTfh and ASC noted over time in Figure 3 is evidence of the resolving immune response in the influenza infected and "control" group. Overall, the rationale for studying this heterogeneous group of patients is not apparent. From data presented later in

Figure 4, it appears that the authors have healthy adult subjects and these could be used to establish the parameters shown in the early parts of the data presented.

2. HLA class I data and IFITM3-SNP could be mentioned in the text but since no relationship was found, these data do not need to be included. The historical data will be unfamiliar to the typical reader and although logical to the authors to evaluate, does not make a contribution to this paper, except to include as a statement in the discussion.

3. Although the block of early cytokines noted by the authors for the data in Figure 1f (IL-6, IL-8 etc), other areas of convergence are also seen (IFN- α , IL-17 IL-10), although slightly less striking and should be discussed.

4. The data in Figure 3 d, e and f present the data as cells per ml does not seem to account for the lymphopenia that is acknowledged and discussed. Additional data on the lymphopenia for each subject should be discussed, presented and commented on, relative to healthy adults as part of the data in Figure 3.

5. The authors' cTfh1 was the main subset of cTfh elicited by infection, (as has been suggested by others studies) but the Figure in 3f is confusing. It appears that correlation between Tfh and ASC shown in panel 3f only correlates total Tfh. The authors state that Figure 3f illustrates cTfh1 but the annotation on the Figure states that these are total cTfh, so perhaps this is simply an error in labeling the Figure. From the supplementary data presented (Suppl. Fig 3), cTfh2 also correlates although more weakly, but these data should be shown in the main paper and commented. Also if the Figure 3f is only the cTfh1, then the total Tfh vs ASC, as well as the other subsets should also be shown as a correlation, also in the main Figure set.

6. One type of data that is lost in Figures such as 3d and 3e is the patterns of individual's abundance of specific subsets of cells at the acute and follow up stage. The authors only present the aggregate patterns. It would be useful for the authors to illustrate the changes over time for each of the subjects, provided as an additional column in panels d and e. This will allow the reader to easily grasp the range in kinetic patterns among the subjects post-infection and if any anomalies are seen, the authors could mine these data with the other accumulated data collected here and comment on it. The same would be useful to do in Figure 4e.

7. The tetramer data are quite nice but for these low numbers, and with the samples available of diverse HLA types, the authors should show and quantify tetramer staining at the acute and follow-up time points in some subjects that are lacking the HLA types that should match the tetramer, ideally some of those that react

Reviewer #3 (Remarks to the Author):

Comments on Nguyen et al for Nature Comm.

This manuscript is a tour de force making a comprehensive analysis of the total immune response to influenza infection, and ILI controls during acute infection and early recovery.

The study is pretty comprehensive including known genetic factors, antibody, B cell, CD4 T cell including cTfh, CD8 T cell and innate immune responses. There is a very rich goldmine of data here that will be invaluable to future studies including provision of a comparator for other infections such as Covid-19. They do a very good job in presenting this and making it all accessible.

I have a few small quibbles that can mostly be corrected easily:

In discussing IFITM3 they should reference the paper Zhang et al which first confirmed the very strong association in a Chinese cohort:

Zhang YH, et al. Interferon-induced transmembrane protein-3 genetic variant rs12252-C is associated with severe influenza in Chinese individuals. Nat Commun. 2013;4:1418. Epub

2013/01/31. doi: 10.1038/ncomms2433. PubMed PMID: 23361009; PMCID: PMC3562464.

They should also reference the paper below by de Jong et al, which first showed they hypercytokinemia in fatal human avian-influenza
de Jong MD, et al. Fatal outcome of human influenza A (H5N1) is associated with high viral load and hypercytokinemia. Nat Med. 2006;12(10):1203-7. Epub 2006/09/12. doi: 10.1038/nm1477. PubMed PMID: 16964257; PMCID: PMC4333202.

As described in the de Jong paper, increased blood levels of the chemokine IP-10 (CXCL10) were the most striking and I wonder why they didn't include this in their analysis here.

The measurement of the CD8 T cell response uses an impressive array of tetramers but even so the levels are rarely above 1% in the blood. It has often been suggested that unlike blood/lymphoid infections such as EBV and HIV where blood levels can be very high (>10% for a single specificity) in acute influenza they may be migrating to the respiratory tract. Do they have any information on that? Comparing BAL lymphocytes might be very revealing.

Also, although they examined well known epitopes presented by common HLA antigens, the immune response is always capable of surprise and they may have missed strong responses to rarer epitopes and HLA presenting molecules.

For the Tfh, is it possible to look at the very small CXCR5+ very high PD1 population in the blood – these are most likely to correspond to the germinal center Tfh although this might not be the case for Th1 type Tfh. They could discuss this.

Overall this is particularly valuable and important work. Of course they can't readily distinguish cause from effect so different hypotheses as to what is going on remain, but future explanations will have to accommodate these data. The authors can address this and the other fairly minor points above in their discussion.

Andrew McMichael

Reviewer #4 (Remarks to the Author):

The manuscript is well-written and articulated. The authors do an exceptional job of defining and assessing numerous immune factors in human patients with mild and severe influenza infections. Overall, this manuscript contains a considerable aggregate of data, and despite the challenges of reporting such involved data, the authors did a commendable job at communicating the findings. The introduction and discussion are well while not overbearing. The methods section could use some enhancement as described below.

Suggestions:

- Please speculating about the differences between H7N9 and seasonal influenza for all of the experiments that they have a comparison.
- page 2 line 65-67: Revise sentence
- line 137- Clarify the acronym AML
- define MAIT cells (function, markers)
- please clarify the number of events collected for flow cytometry
- Figure 1d: Please flip the x- and y-axis. Please separate 'seasonal flu' patients into recovered and died
- Figure 1i: please add statistics
- Figure 1 overall: Please add data for H7N9 for all of the graphs or move the H7N9 data all into the supplemental figures.
- Figure 2e: Michigan/15 is included on the H1N1 YAM graph while Cal/09 is included on every other graph

- Figure 3h: clarify why Cal/09 and Swi/13 were used for the H1N1 graphs while Cal/09 and HK/14 used for the H3N2 graphs
- Figure 5f: clarify why were the flu- patients only tested with M1
- Figure 5j: Remove the N=1 columns as they are misleading when compared to the N=4, N=19, and N=24 columns
- Figure 6a and b: Please have the cytokines in the same order for both of the figures

REVIEWER COMMENTS

We thank immensely the Reviewers for their comments and for acknowledging the novel aspects of our study.

Reviewer #1 (Remarks to the Author):

I commend the authors on their efforts here to explore the immunopathogenesis of influenza infection and find this manuscript very interesting and useful.

We appreciate the Reviewer's comments.

Comments:

1. I would like to see some discussion of the weaknesses of the study. In particular the authors should discuss the fact that they are only observing FLU+ patients who were sick enough to present for testing/care. This is a skewed population and represents the tip of the iceberg in terms of influenza cases worldwide. Some of their finding may not be applicable to the less ill individuals or individuals at less risk of complication/ more severe illness such as very young healthy individuals. Also there should be a discussion on how this study was done overall on a very small number of people and it is unclear to me how diverse these individuals were in terms of age, sex, race, etc.

We have included a paragraph in Discussion (page 10 and below) to discuss the limitations of the study, as per Reviewer's comments, and the smaller cohort size. The age range is depicted in Figure 1c. We have also summarised the demographics of the DISI cohort in Supplementary Table 1.

“There are limitations to this study. The DISI cohort included Flu+ patients who were ill enough to be admitted to hospital, thus representing a skewed population of severe influenza cases. The majority of these patients were also already at high-risk, with 86% having pre-existing comorbidities. Therefore, some findings of our study may not be applicable to the general population that experience mild influenza symptoms, have no comorbidities or are very young healthy individuals. We acknowledge that our DISI study included a small cohort (n=65: n=45 Flu+ and n=20 Flu-), but we performed a comprehensive study, which included longitudinal sampling during the hospital stay and a follow-up sample at ~30 days after discharge. Although we cannot readily distinguish cause from effect, different hypotheses on the immune mechanisms underlying influenza disease severity remain. However, our DISI study provides a valuable and important body of work that provides key insights into understanding immune responses during severe influenza disease. Furthermore, as besides the SHIVERS report²⁹, there are no other influenza studies that have analyzed paired acute and convalescent influenza samples, our study provides important data on the specific interplay between host genetics, innate and adaptive immune responses”.

2. It would be useful to have some more information about the patients they enrolled in terms of demographics, etc.

We have summarised the patient demographics in Supplementary Table 1 and also provide their individual demographic and HLA information in Supplementary Table 2.

3. The statement that their work supports The Who recommendations for vaccination should be changed or removed. They looked at a population of individuals who presented for care and the ones who were flu positive were less likely to be vaccinated, but the flu negatives were also presenting for care. There are multiple confounding factors here. First is that there is no way to know for sure that the FLU negative did not have influenza, but the window for detection was missed. The skewed population they looked at may not represent the general population at

large, i.e. healthy individuals, children, and so one, and so one must be careful here. They should either remove it or simply state that in this particular population of people presenting to hospital it suggests that vaccination may be beneficial.

We agree with the Reviewer and have removed these statements in the Discussion on page 9.

4. Lastly I think it would be helpful if the authors could better summarize their conclusions from all of their work as they end the discussion to highlight the key points of their findings. As it is I feel quite a bit of the information gets lost in the discussion and may cause the findings to not be appreciated for what they are.

We thank the Reviewer for the comments and have re-written the start of Discussion (page 8) to better summarise the key findings:

“We describe the broadest to-date immune cellular networks, in association with clinical and genetic characterization, underlying recovery from influenza infection, which is highly relevant to other infectious diseases. We provide the first evidence that the breadth of robust immune responses to influenza viruses can be measured in peripheral blood prior to patient recovery. Firstly, activated Tfh cells emerged in the blood in parallel with ASCs, both peaking between days 7-10 after disease onset in both Flu+ and Flu- groups, prior to patients’ recovery. This is the first evidence of Tfh cells in the blood during and following influenza virus infection. Secondly, hospitalized influenza-patients had increased IL-6, IL-8, MIP-1 α/β and IFN- γ inflammatory cytokines, which strongly positively correlated with each other and were associated with lower influenza-specific responses from CD8⁺ T cells, CD4⁺ T cells, NK cells, MAIT and $\gamma\delta$ T cells, and lower ASC and Tfh responses. This is the most extensive study covering innate cells, adaptive cells and 17 cytokines/chemokines. Thirdly, influenza-specific B cell responses positively correlated with ASC and Tfh responses, but differed to vaccination-induced B-cell responses, revealing contrasting phenotype and isotype characteristics that will inform future B cell-based vaccine designs to promote both IgG⁺ and IgA⁺ responses. Fourthly, patient-specific peptide/MHC-tetramer staining captured the broadest array of epitope-specific CD8⁺ and CD4⁺ T cells during acute IAV infection to date in one study, by utilizing an extensive range (n=21) of peptide/MHC class-I and class-II tetramers. These tetramers cover the most frequent HLA alleles, with an estimated population coverage of 63-100% across all ethnicities, and provide essential tools for future T cell-immune monitoring of newly-emerging influenza viruses. Lastly, lower influenza-specific $\gamma\delta$ T and MAIT cell responses were associated with more severe patients with higher SOFA scores, but not for NK cells, CD4⁺ and CD8⁺ T cells, showing that early influenza-specific NK, CD4⁺ and CD8⁺ T cell responses were important in driving patients’ recovery from influenza disease.”

Reviewer #2 (Remarks to the Author):

This is wonderfully rich data set, but it could be presented in a way that permits more patterns exhibited by individual subjects to be apparent. Most data are presented in aggregate and although the last Figure analyzes them on a subject-to-subject basis, it will be difficult for the reader to extract specific features of the immune response to infection that they might be interested in. It is recommended that the paper focus on key issues that can be presented in a more centered conceptual framework, allowing more space for discussion and re-analyses of key data and place the ancillary issues and data into the supplemental data, if they desire. For example, the antibody data and lineage relationships in Figure 2 do not appear to be essential to the main issues addressed in this study, nor do the SNP data in Figure 1.

We thank the Reviewer for their comments. Following the Reviewer's suggestions here and below (point #2), we have removed the SNP and HLA data from Figure 1 and have mentioned them in Discussion. However, we would like to keep the antibody data in Figure 2, as we then discuss how the antibodies relate to Tfh, ASC and influenza-specific B cells in the next Figure.

Recommendations

1. The choice of the major control group seems problematic. Most are pathogen-infected and several are diagnosed with RSV or parainfluenza virus. These infections would have unique and not insignificant distinctive features of induced immunity. The logic for this is not clear and would seem to derive more out of ease of sampling subjects in parallel. Data for most parameters should be presented for a similarly sized cohort of healthy controls subjects that are age and comorbidities-matched. The changes in cTfh and ASC noted over time in Figure 3 is evidence of the resolving immune response in the influenza infected and "control" group. Overall, the rationale for studying this heterogeneous group of patients is not apparent. From data presented later in Figure 4, it appears that the authors have healthy adult subjects and these could be used to establish the parameters shown in the early parts of the data presented.

We agree with the Reviewer that our Flu- patient group is not the typical "control" group, such as a healthy group, and agree that they provide another unique cohort to investigate cTfh and ASC responses in other respiratory virus infections, which we found were also dynamic. We have made several changes within the entire manuscript to *not* address the Flu- group as a control group, but a unique cohort of patients with other respiratory illnesses including some with known respiratory viral infection.

We have previously described cTfh and ASC responses using whole blood staining in healthy individuals prior to and following influenza vaccination (Koutsakos *et al.* STM 2018 PMID: 29444980), therefore we did not include any healthy data here for cTfh and ASC responses. To improve the manuscript, we have made further comparisons between the levels of cTfh and ASC in our patient cohort with our previously described healthy cohort within the Results section (see below, page 5) and added a median healthy baseline level in Figures 3e and 3f (see below) to highlight these differences.

"The number of ASCs was significantly higher (~2-8 median fold) during acute infection compared to follow-up for both Flu+ and Flu- groups (Fig. 3e). Moreover, Flu+ patients at acute phase had higher ASC numbers than our previously described healthy cohort prior to vaccination (4.4 median fold) and at d7 post vaccination (2.6 median fold), the peak of the vaccine response. Activated cTfh1 cells were also trending higher at acute compared to follow-up (~2 fold, p=0.0519, Mann-Whitney test, Fig. 3f). Flu+ patients at acute had 1.5 median fold higher levels of activated cTfh1 cells compared to healthy individuals at baseline, but were very comparable at d7 post vaccination in the healthy donors."

New Figure 3e and 3f (previously 3d and 3e).

Since these assays require fresh whole blood assays, we regrettably cannot retrospectively recruit any new donors that are age-matched with comorbidities around the time the DISI cohort was established (2014-2017).

2. HLA class I data and IFITM3-SNP could be mentioned in the text but since no relationship was found, these data do not need to be included. The historical data will be unfamiliar to the typical reader and although logical to the authors to evaluate, does not make a contribution to this paper, except to include as a statement in the discussion.

We totally agree with the Reviewer and have removed the SNP and HLA data from Figure 1 and have mentioned them in Discussion.

3. Although the block of early cytokines noted by the authors for the data in Figure 1f (IL-6, IL-8 etc), other areas of convergence are also seen (IFN- α , IL-17 IL-10), although slightly less striking and should be discussed.

We have highlighted this important observation in the Results section (page 4, “as well as the convergence of an IFN- α , IL-17 and IL-10 cluster”) and Discussion section (page 9, “in addition to the emergence of an IFN- α , IL-17 and IL-10 cluster”).

4. The data in Figure 3 d, e and f present the data as cells per ml does not seem to account for the lymphopenia that is acknowledged and discussed. Additional data on the lymphopenia for each subject should be discussed, presented and commented on, relative to healthy adults as part of the data in Figure 3.

We thank the Reviewer for these insightful suggestions. We have acknowledged and discussed the patients’ lymphopenia at the acute phase in the Results section (see below, page 5) and have now included new data, consisting of cell counts for CD45⁺ lymphocytes and each major cell subset (CD3⁺, CD8⁺ and CD4⁺ T cells, CD16/56⁺ NK cells and CD19⁺ B cells) for Flu+ and Flu- patients in comparison to healthy individuals as a new Supplementary Figure 3 (please see below).

“The number of activated cTfh1 cells for acute Flu+ patients was only 1.5 median fold higher compared to healthy individuals at baseline, and was slightly lower than d7 post vaccination in the healthy controls. This could be in part due to the patients’ lymphopenic state during acute illness where the total CD45⁺ lymphocyte and subset-specific cell counts were significantly lower at acute-V1 compared to follow-up, except for B cells, which explains the higher ASC numbers observed in acute patients. Cell counts were less variable in the Flu-patients and more stable in healthy donors (Supplementary Fig. 3a-c).”

New Supplementary Figure 3. Lymphopenia observed in patients during acute infection. (a,b), Absolute numbers of cell subsets at acute (V1) and matching follow-up time-points in Flu+ and Flu- patients, where available. Statistical significance ($p < 0.05$) was determined using Wilcoxon test between acute and follow-up. c, Absolute numbers of cell subsets at acute (V1) and follow-up time-points in Flu+ patients in comparison to Flu- and healthy donors from the 2015-2016 pre-vaccinated cohort. Median, IQR and exact n numbers are shown for each group. NS = not significant.

5. The authors' cTfh1 was the main subset of cTfh elicited by infection, (as has been suggested by others studies) but the Figure in 3f is confusing. It appears that correlation between Tfh and ASC shown in panel 3f only correlates total Tfh. The authors state that Figure 3f illustrates cTfh1 but the annotation on the Figure states that these are total cTfh, so perhaps this is simply an error in labeling the Figure. From the supplementary data presented (Suppl. Fig 3), cTfh2 also correlates although more weakly, but these data should be shown in the main paper and commented. Also if the Figure 3f is only the cTfh1, then the total Tfh vs ASC, as well as the other subsets should also be shown as a correlation, also in the main Figure set.

We thank the Reviewer and apologise for the labelling error. Figure 3f is indeed a correlation between activated cTfh1 cells and ASCs. We have now split Figure 3 into two figures to add all the correlation graphs, including total cTfh vs ASCs, into the main Figure 3 (please see below) and further described in the Results section (page 5), as shown below. We really appreciate the Reviewer's suggestions to improve the manuscript.

“cTfh1 responses strongly correlated with ASC responses ($r_s = 0.7060$, $p < 0.0001$) during acute influenza virus infection, but less so for total cTfh ($r_s = 0.5397$, $p = 0.0003$) and cTfh2 cells ($r_s = 0.3741$, $p = 0.0174$) and no correlation with cTfh17 subsets (Fig. 3g). Whereas strong correlations were observed between acute ASC responses and total cTfh, cTfh1 and cTfh2 T cell subsets for Flu- patients (Fig. 3h).”

New Figure 3g and 3h. Correlations of cTfh subsets with ASC during acute infection.

6. One type of data that is lost in Figures such as 3d and 3e is the patterns of individual's abundance of specific subsets of cells at the acute and follow up stage. The authors only present the aggregate patterns. It would be useful for the authors to illustrate the changes over time for each of the subjects, provided as an additional column in panels d and e. This will allow the reader to easily grasp the range in kinetic patterns among the subjects post-infection and if any anomalies are seen, the authors could mine these data with the other accumulated data collected here and comment on it. The same would be useful to do in Figure 4e.

We have plotted individual patient datapoints with connecting lines for longitudinal samples of ASC and cTfh1 subsets to show the patients' dynamic responses between Flu+ and Flu- patients. This is now incorporated within the Results section (page 5) and as a new Figure 3d (below).

New Figure 3d.

We have not done the same analysis for Figure 4e (New Figure 5e) relating to the frequency of total cytotoxic molecules for NK cells, MAIT cells and CD4⁺ and CD8⁺ T cells, as we believe it is not the main focus of the study in terms of our key findings.

7. The tetramer data are quite nice but for these low numbers, and with the samples available of diverse HLA types, the authors should show and quantify tetramer staining at the acute and follow-up time points in some subjects that are lacking the HLA types that should match the tetramer, ideally some of those that react.

We thank the Reviewer for the comment. We have over a decade of experience in tetramer staining and TAME of high and low numbers of T cells (enriched and unenriched, effector,

memory and naive) in the blood and tissues (spleens, lymph nodes, lungs, cord blood) of healthy donors and influenza patients, with our generated library of well-validated peptide/HLA tetramers (e.g. Koutsakos *Nat Immunol* 2019, van de Sandt *Nat Commun* 2019, Sant *Plos Path* 2020, Sant *Front Immunol* 2018, Pizzola *J Clin Invest* 2018, Nguyen *J Leu Biol* 2018, Valkenburg *PNAS* 2016). We have successfully sorted single-cells for RNAseq and paired TCR $\alpha\beta$ analysis using our tetramer enrichment methods and are very confident of our tetramer staining. We detected no tetramer⁺ T cells in any of the flow through fractions. These tetramers have all been validated by staining T cell lines generated from healthy buffy packs. We show the validation of the new DR1/4/11-HA tetramers in healthy T cell lines, since the tetramer⁺ populations were low in some patients (**Rebuttal Fig. 1**). However, tetramer staining is much cleaner in *ex vivo* TAME compared to expanded T cell lines. Unfortunately, we do not have sufficient vials left that have matched acute and follow-up samples to perform negative-tetramer staining. We hope this is acceptable for the Reviewer.

Rebuttal Figure 1. Validation of class II DR-HA tetramers (epitope restricted by HLA-DR*01:01; 4:01 and 11:01). PBMCs from healthy donors were expanded for 10 days with HA₃₂₂₋₃₃₄ peptide derived from H3N2 viruses before (top panels) and since 2012 (bottom panels) and then stained with the cognate peptide/HLA class II tetramer. Cells were gated on live CD3⁺ T cells.

Reviewer #3 (Remarks to the Author):

Comments on Nguyen et al for Nature Comm.

This manuscript is a tour de force making a comprehensive analysis of the total immune response to influenza infection, and ILI controls during acute infection and early recovery.

The study is pretty comprehensive including known genetic factors, antibody, B cell, CD4 T cell including cTfh, CD8 T cell and innate immune responses. There is a very rich goldmine of data here that will be invaluable to future studies including provision of a comparator for other infections such as Covid-19. They do a very good job in presenting this and making it all accessible.

We thank the Reviewer for their generous and positive comments.

I have a few small quibbles that can mostly be corrected easily:

In discussing IFITM3 they should reference the paper Zhang et al which first confirmed the very strong association in a Chinese cohort:

Zhang YH, et al. Interferon-induced transmembrane protein-3 genetic variant rs12252-C is associated with severe influenza in Chinese individuals. *Nat Commun.* 2013;4:1418. Epub

2013/01/31. doi: 10.1038/ncomms2433. PubMed PMID: 23361009; PMCID: PMC3562464. They should also reference the paper below by de Jong et al, which first showed they hypercytokinemia in fatal human avian-influenza de Jong MD, et al. Fatal outcome of human influenza A (H5N1) is associated with high viral load and hypercytokinemia. Nat Med. 2006;12(10):1203-7. Epub 2006/09/12. doi: 10.1038/nm1477. PubMed PMID: 16964257; PMCID: PMC4333202.

We have now referenced these papers in the Introduction (page 2) and in the Discussion (page 9).

As described in the de Jong paper, increased blood levels of the chemokine IP-10 (CXCL10) were the most striking and I wonder why they didn't include this in their analysis here.

We would have loved to include IP-10, but unfortunately this was not included in our cytokine/chemokine panel design. We do have IFN- γ which is an inducer of IP-10 and was strongly correlated with IL-6, IL-8, MIP-1 α and MIP-1 β at the acute phase but not at convalescence, which is now highlighted in the Results (page 4) and Discussion sections (page 9).

The measurement of the CD8 T cell response uses an impressive array of tetramers but even so the levels are rarely above 1% in the blood. It has often been suggested that unlike blood/lymphoid infections such as EBV and HIV where blood levels can be very high (>10% for a single specificity) in acute influenza they may be migrating to the respiratory tract. Do they have any information on that? Comparing BAL lymphocytes might be very revealing.

This is a very intriguing thought but unfortunately, we do not have access to BAL lymphocytes from acute influenza patients. However, we have previously stained human lung cells from deceased organ donors with influenza class I tetramers (see Fig. 4e-g below from Pizzolla, Nguyen, Sant *et al.* 2018, J Clin Invest, PMID: 29309047) and observed high frequencies of influenza-specific CD8⁺ T cells within the memory CD45RO⁺CD8⁺ T cell population, up to 8%. It is thus tempting to speculate that perhaps these CD8⁺ T cell populations in the lung may expand to even greater numbers during acute infection.

[Redacted]

Pizzolla *et al.* 2018 Figure 4
J Clin Invest. 2018;128(2):721-733. <https://doi.org/10.1172/JCI96957>.

We have now inserted the Reviewer's suggestions into the Discussion section (page 10) as follows:

"Intriguingly, influenza-specific T cells were rarely above 1% of total CD8⁺ T cells in the blood and were not significantly increased during acute infection compared to the convalescence phase. This is in stark contrast to blood and lymphoid infections such as EBV and HIV-1 where blood levels can be very high (>10% for a single specificity). It is tempting to speculate that perhaps they may be migrating to the respiratory tract during acute influenza infection, although we do not have BAL samples from Flu⁺ patients to test this hypothesis. However, we have previously stained human lung cells from deceased organ donors with influenza class I tetramers (HLA-A2-M1₅₈ and HLA-B57-NP₁₉₉) and observed high frequencies

of up to 8% of influenza-specific CD8⁺ T cells within the memory CD45RO⁺CD8⁺ T cell population⁴⁷. We speculate that perhaps these cell populations in the lung may expand to even greater numbers during acute infection.”

Also, although they examined well known epitopes presented by common HLA antigens, the immune response is always capable of surprise and they may have missed strong responses to rarer epitopes and HLA presenting molecules.

We have acknowledged this in the Discussion (page 10) as follows:

“Although we may have missed strong responses to rarer epitopes and HLA presenting molecules, our results represent the broadest array of influenza-specific tetramer⁺ T cell responses during acute influenza infection...”

For the Tfh, is it possible to look at the very small CXCR5⁺ very high PD1 population in the blood – these are most likely to correspond to the germinal center Tfh although this might not be the case for Th1 type Tfh. They could discuss this.

We analysed CXCR5⁺PD-1^{high} expression on total CD4⁺ T cells in a small subset of Flu+ patients and found very low levels of these CXCR5⁺PD-1^{high} cTfh cells in the blood at both acute and follow-up time-points (**Rebuttal Fig. 2**). This is in contrast to a more prominent population found in tonsils (Brenna *et al.* Cell Rep 2020, PMID: 31914381) and lymph nodes (Hill *et al.* JEM 2019, PMID: 31175140). We have included our observations in the Discussion (page 9) as follows:

“A very small subset of CXCR5⁺PD-1^{high} cTfh cells in the blood have been shown to correspond to CXCR5⁺PD-1^{high} germinal centre Tfh cells by sharing transcriptional and TCR repertoire profiles^{41,42}. We analyzed CXCR5⁺PD-1^{high} expression on total CD4⁺ T cells in a small subset of Flu+ patients and also found very low levels in the blood at both acute and follow-up time-points (0-0.2%, data not shown). This is in contrast to a more prominent population found in lymph nodes⁴¹ and tonsils⁴².”

Rebuttal Figure 2. CXCR5⁺PD-1^{high} cTfh cells in the blood. Representative CXCR5 and PD-1^{high} expression and frequencies of CXCR5⁺PD-1^{high} cTfh cells of total CD4⁺ T cells in Flu+ patients at acute and follow-up (F_{up}). Cells were gated on live/CD3⁺/CD4⁺ T cells.

Overall this is particularly valuable and important work. Of course they can't readily distinguish cause from effect so different hypotheses as to what is going on remain, but future explanations will have to accommodate these data. The authors can address this and the other fairly minor points above in their discussion.

We have addressed this as a limitation to the study (see below, Discussion page 11) and have addressed all the above points in the Discussion as indicated in the page numbers and tracked changes.

“Although we cannot readily distinguish cause from effect, different hypotheses on the immune mechanisms underlying influenza disease severity remain. However, our DISI study provides a valuable and important body of work that provides key insights into understanding immune responses during severe influenza disease.”

Andrew McMichael

We thank Professor McMichael immensely for his positive and valuable comments.

Reviewer #4 (Remarks to the Author):

The manuscript is well-written and articulated. The authors do an exceptional job of defining and assessing numerous immune factors in human patients with mild and severe influenza infections. Overall, this manuscript contains a considerable aggregate of data, and despite the challenges of reporting such involved data, the authors did a commendable job at communicating the findings. The introduction and discussion are well while not overbearing. The methods section could use some enhancement as described below.

We thank the Reviewer for their generous and positive comments.

Suggestions:

- Please speculating about the differences between H7N9 and seasonal influenza for all of the experiments that they have a comparison.

We have compared and contrasted H7N9 and seasonal influenza immune responses in the cytokine analysis, IFN- γ functional assay and HLA-DR/CD38 expression on T cells within the Results and Discussion, which is all the comparisons available for H7N9.

- page 2 line 65-67: Revise sentence

We have revised as: “Here, we describe the broadest to-date immune cellular networks underlying recovery from influenza infection *which are* highly relevant to other infectious diseases.”

- line 137- Clarify the acronym AML

We have spelt out AML as “acute myeloid leukemia”.

- define MAIT cells (function, markers)

We have defined MAIT cells as IFN- γ -producing innate cells expressing CD161⁺TRAV1-2⁺, which was validated by the MR1-5'OP-RU-tetramer in 51% of samples, in the legend of Figure 4 and Supplementary Fig. 5c,d.

- please clarify the number of events collected for flow cytometry

We have acquired all events for most of the assays and have included event details in the Methods section (page 12).

- Figure 1d: Please flip the x- and y-axis. Please separate ‘seasonal flu’ patients into recovered and died

We thank the Reviewer for this comment and have modified the figure accordingly, as shown below.

New Figure 1d.

• Figure 1i: please add statistics

Statistics to Fig. 1i (which is now Fig. 1e) cannot be applied because it is a representative figure showing one individual's cytokine profile per pie/bar graph for each severity group.

• Figure 1 overall: Please add data for H7N9 for all of the graphs or move the H7N9 data all into the supplemental figures.

We have included relevant H7N9 data to easily compare and contrast the level of disease severity by the number of days in hospital and the different cytokine profiles from our previous publication, which has already been described in detail (Wang *et al.* 2013 PNAS, PMID: 24367104). We believe this is an effective and visual way to compare H7N9 and seasonal influenza and would like to leave it in the main Fig. 1.

• Figure 2e: Michigan/15 is included on the H1N1 YAM graph while Cal/09 is included on every other graph

We thank the Reviewer for this comment and have replaced Michigan/15 to Cal/07 so that they are consistent.

• Figure 3h: clarify why Cal/09 and Swi/13 were used for the H1N1 graphs while Cal/09 and HK/14 used for the H3N2 graphs

Patient #16 was recruited in 2014, while patient #48 was recruited in 2016. There were major clade changes in seasonal H3N2 viruses dominating each season and so we have chosen H3 probes that closely matched the strains that were predominantly infecting our patients for that year. We have clarified this in the Methods (page 12): “Flu+ patients were stained with recombinant (r) HA probes from H1 (California/7/2009), H3 (Switzerland/9715293/2013 *before 2016* or Hong Kong/4801/2014 *since 2016*)...”

• Figure 5f: clarify why were the flu- patients only tested with M1

Due to the limited number of Flu- patients enrolled, they represent a unique set of patient cells with other respiratory illness, thus we made a decision to focus on only testing the main and well characterised immunodominant A2/M1₅₈ tetramer-specific CD8⁺ T cell responses in Flu- patients as a representative comparison in terms of numbers and activation profiles.

• Figure 5j: Remove the N=1 columns as they are misleading when compared to the N=4, N=19, and N=24 columns

We would like to show the two acute phenotypes individually, similar to Fig. 5i (which is now Fig. 6i), rather than pooling them together and inaccurately describing the mean and SD on 2 data points.

• Figure 6a and b: Please have the cytokines in the same order for both of the figures

These heat maps (now Fig. 7a,b) are based on unsupervised clustering and so the order of the labels are ordered unbiasedly.

REVIEWERS' COMMENTS

Reviewer #1 (Remarks to the Author):

I feel my comments have been adequately addressed by the authors.

Reviewer #2 (Remarks to the Author):

The authors have made considerable effort to make the studies more focused and accessible to the reader.

Reviewer #3 (Remarks to the Author):

The authors have responded well to my comments and to those of the other referees, making appropriate changes to the manuscript.

Reviewer #4 (Remarks to the Author):

The revised manuscript is acceptable addressing all concerns of this reviewer.